# Evolutionary modelling indicates that mosquito metabolism shapes the life-history strategies of *Plasmodium* parasites

Paola Carrillo-Bustamante [1] ✉, Giulia Costa [1], Lena Lampe[1,2] & Elena A. Levashina [1] ✉

Within-host survival and between-host transmission are key life-history traits of single-celled malaria parasites. Understanding the evolutionary forces that shape these traits is crucial to predict malaria epidemiology, drug resistance, and virulence. However, very little is known about how *Plasmodium* parasites adapt to their mosquito vectors. Here, we examine the evolution of the time *Plasmodium* parasites require to develop within the vector (extrinsic incubation period) with an individual-based model of malaria transmission that includes mosquito metabolism. Specifically, we model the metabolic cascade of resource allocation induced by blood-feeding, as well as the influence of multiple blood meals on parasite development. Our model predicts that successful vector-to-human transmission events are rare, and are caused by long-lived mosquitoes. Importantly, our results show that the life-history strategies of malaria parasites depend on the mosquito's metabolic status. In our model, additional resources provided by multiple blood meals lead to selection for parasites with slow or intermediate developmental time. These results challenge the current assumption that evolution favors fast developing parasites to maximize their chances to complete their within-mosquito life cycle. We propose that the long sporogonic cycle observed for *Plasmodium* is not a constraint but rather an adaptation to increase transmission potential.

Malaria, a parasitic disease caused by *Plasmodium spp*, poses one of the greatest medical and economical challenges in our society. *Plasmodium* parasites infected 229 million people in 2019 alone and claimed the lives of 409,000 individuals[1]. Malaria parasites exhibit a complex life cycle, invading and developing in a wide range of host environments, both within blood-feeding mosquitoes (their definitive host) and in vertebrates (their intermediate hosts). For malaria parasites, within-host survival and transmission between hosts are major components of fitness, consequently natural selection is expected to benefit adaptations that maximize these components. Understanding the mechanisms within each host that affect parasite's life-history traits, and how parasites adapt to changes in their environment is

crucial to determine potential bottlenecks in transmission, and predict the course of malaria epidemics.

Given the parasite's complex life cycle, the study of *Plasmodium* evolution is not trivial. Important insights have been obtained by applying ecological and evolutionary theories to study some reproductive strategies utilized by malaria parasites in the vertebrate host[2–7]. Here, malaria parasites face life-history trade-offs typical to all sexually reproducing organisms, when resources must be divided between growth (asexual replication in red blood cells) and reproduction (production of non-replicating sexual stages, gametocytes)[8]. Malaria parasites indeed deploy several strategies to alter their conversion rate (investment between asexuals and gametocytes)[3,5,6,9], and sex

[1]Vector Biology Unit, Max Planck Institute for Infection Biology, 10117 Berlin, Germany. [2]Present address: Physiology and Metabolism Laboratory, The Francis Crick Institute, NW11AT London, UK. ✉e-mail: carrillo@mpiib-berlin.mpg.de; levashina@mpiib-berlin.mpg.de

allocation (investment into female vs male gametes)[10,11] to adapt to changes in their host environment[12], demonstrating a high level of adaptive phenotypic plasticity.

Although the mosquito is *Plasmodium* definitive host, the evolution of malaria life-history traits in the vector remains largely unexplored, and trade-offs are expected to occur here as well. For transmission to be successful, the within-vector development period must be shorter than the mosquito's average life span. Given the assumed short life-span of mosquitoes in the field, the fitness of malaria parasites is thought to increase with a short sporogonic period. Yet, a parasite developing too fast might not accumulate sufficient resources, which could limit the quantity and/or quality of produced sporozoites—its transmissible form. The trade-off between within-vector survival and between-hosts transmission might be further affected by numerous aspects of mosquito biology that have a direct effect on parasite development. For example, the acquisition of multiple blood meals increases oocyst size[13], oocyst growth rates[14–16], and sporozoite numbers[17], possibly contributing to a rapid yet successful parasite development.

Similarly, the metabolic status of the mosquito is crucial for *Plasmodium* development. *Anopheles* mosquitoes mostly feed on plant nectar, and only females take a blood meal to replenish their amino acid and lipid stores required for their reproductive cycle. In contrast to non-hemophagous insects like *Drosophila* that progressively accumulate resources for reproduction, mosquitoes have evolved a different strategy, characterized by a rapid nutrient assimilation from a blood meal. This process floods the mosquito with the essential nutrients for egg development within a short period of 48 hours, after which the reproductive investment is restrained[18]. *Plasmodium* parasites exploit the blood-feeding behavior as an entry point and also as nutritional source for their own development. The co-dependance on blood meal derived lipids for egg and parasite development has been reported to rely on nutrients/lipids carried by the lipid transporter lipophorin[14,19]. Interestingly, the allocation of nutrients to reproduction inversely affects oocyst and sporozoite development. For example, increasing reproductive investment by prolonging the mosquito reproductive cycle, compromises *Plasmodium*'s sporogonic development. Consequently, a restriction of reproductive investment benefits parasite development[18]. Importantly, mosquitoes do not exhaust all available resources on reproduction[18], and therefore, the surplus metabolic resources that are available to parasites can vary from one mosquito to another.

During the typically studied single blood meal scenario, parasites are thought to interact non-competitively with their vector, as they scavenge the surplus internal resources only after reproductive investment is restrained and successful oviposition occurred[14,18–20]. However, whether this symbiotic interaction is relevant for natural settings of multiple feedings remains unexplored and is difficult to study experimentally. Here, we present a theoretical framework that integrates the effects of within-vector metabolism (i.e. the metabolically induced resource allocation initiated by blood feeding) into an individual-based model of malaria transmission, with the aim to answer the following questions: (1) how does mosquito feeding behavior and metabolism affect *Plasmodium* development? (2) how does mosquito metabolism shape the evolution of *Plasmodium* life-history traits?

We show that malaria parasites exploit a small proportion of the mosquito population for transmission: rare mosquitoes that are long lived and take multiple blood meals during their life span. Moreover, our mosquito metabolism model demonstrates that malaria parasites compete for metabolic resources within their vector and benefit from the nutrients acquired after the second blood meal at the expense of mosquito reproduction. Importantly, we find that our model selects for parasites with longer sporogony time to maximize transmission potential. The evolution of long sporogony time critically depends on

mosquito metabolism: when we let parasites evolve without considering mosquito metabolism, we observe that short developmental times are instead selected. We therefore conclude that mosquito metabolism profoundly affects the evolution of *Plasmodium* parasites, offering a new perspective for understanding malaria epidemiology and transmission.

## Results

### Multiple blood meals result in competitive parasite-vector interactions

We first studied the effect of multiple (>2) blood meals on *Plasmodium* development within its mosquito host with a simple mathematical model that focuses on how nutrients are allocated after the ingestion of a blood meal inside the mosquito host (Fig. 1a). The acquisition of a blood meal triggers essential metabolic pathways that ensure a robust egg development within three days. The major physiological event is vitellogenesis, a process finely orchestrated by the steroid hormone 20-hydroxyecdysone (20E)[21–24], during which essential nutrient transporters are secreted by the insect's fat body, and transferred to the ovaries where they provide nutrients for growing eggs[25]. Here we abstract these complex processes and develop a model in which the within-host energy reserves of a female mosquito ($R$) grow after the ingestion of a blood meal ($\sigma(t)_{BM}$), activating the necessary signal for the steroid hormone 20E synthesis in the ovaries, here represented with a blood meal-activated variable $\beta_E(t)$. Once activated, a proportion of the host resources are mobilized into the fat body for the production of the necessary yolk proteins, which are then utilized by developing oocytes ($E$) in the ovaries. The remaining host resources are invested into other physiological processes, including immunity, physical activity, and waste, at a rate $\delta_R$. If a blood meal is infected with *Plasmodium*, blood-borne sexual forms of the parasite fuse and convert into motile ookinetes, which in turn traverse the midgut epithelium and by day three after infection round-up to form oocysts. Mature oocysts generate then thousands of infective-to-human sporozoites that ultimately accumulate in the salivary glands. The duration of this parasite developmental process is known as extrinsic incubation period (EIP) (hereafter called sporogonic cycle $T_{sp}$), and lasts approximately 10-14 days in natural systems[26]. We simplify this complex process and model oocysts ($O$) that grow by accumulating resources taken from the vector's reserves at a constant rate $\beta_P$. Importantly, the ookinetes and young oocysts are less active metabolically and require approximately 5-6 days before initiating active growth[19]. We model these early metabolically inactive parasites by having a very low initial metabolic energy in the oocyst compartment ($O(0) = 0.01$). Once mature, oocysts transfer their internal energy into sporozoites ($S$) at a rate $\gamma(t)$. The model consists of a system of ordinary differential equations (ODE). See Methods for details.

As a control, we first simulated the allocation of internal energy resources after a mosquito has acquired one non-infectious (BM) and one infectious blood meal (iBM) (Fig. 1b, left columns). After ingestion of a blood meal, the mosquito's internal reserves are directed to the energy compartment used by reproduction, resulting in the rapid development of eggs. Importantly, the dynamics of reproduction are not affected in the presence of *Plasmodium*. Because in our model the total rate of energy acquisition grows with increasing amount of metabolic energy inside the oocyst compartment, the early (and metabolically low energy) parasites cannot scavenge enough resources, resulting in a very weak competition with its vector, like that observed experimentally[14,19,20]. We next simulated the acquisition of three blood meals (Fig. 1b, right columns). Interestingly, the energy accumulated by developing eggs during the second and third gonotrophic cycle decreases only in infected mosquitoes. Because the mosquito's second and third reproductive cycles and oocysts development now occur simultaneously, and the oocysts are sufficiently large to scavenge more nutrients, there is large competition for resources.

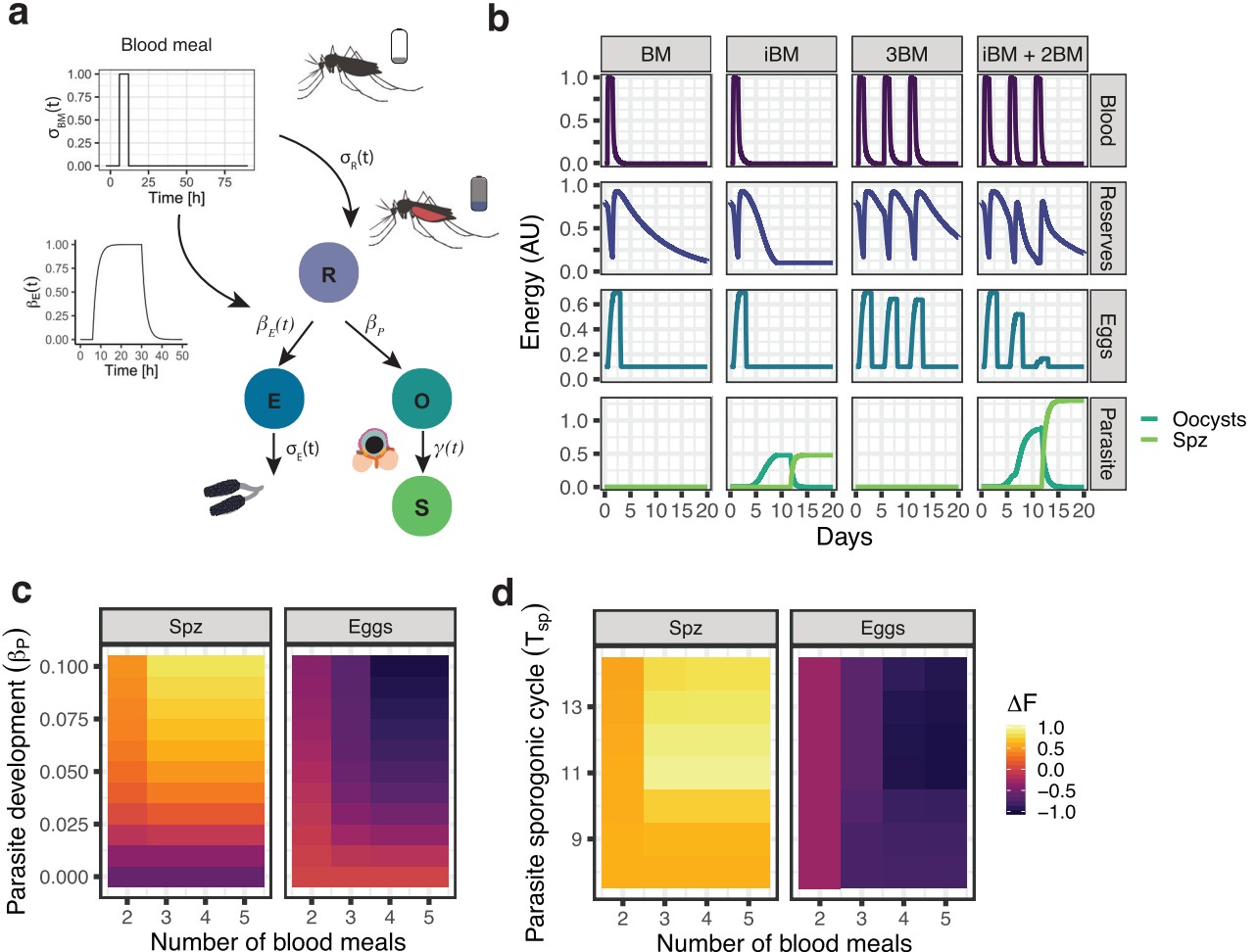

**Fig. 1 | The parasite interacts competitively with its mosquito host after the second blood meal. a** Schematic representation of the within-vector model of metabolic resource allocation. A successful blood meal replenishes the initial energy resources, a small proportion of which will be used for the development of eggs and *Plasmodium* parasites. The nutrient mobilization is modeled considering a periodic blood meal behavior given by $\sigma(t)_{BM}$ and four different compartments: within-host energy reserves ($R$), energy invested in reproduction ($E$), developing oocysts ($O$), and sporozoites ($S$). The full model is described by Eqs. (3)-(7) and the parameters are given in Table 1. **b** Simulation of internal energy resources after one (BM / iBM) and three (3BM/ iBM + 2BM) blood meals (depicted in purple). The ingestion of one blood meal (BM) activates the mobilization of mosquito resources (depicted in dark blue) to the ovaries, where energy accumulates in developing oocytes (depicted in light blue). The investment in reproduction remains unaffected during an infectious blood meal (iBM). After successful oviposition, the remaining internal reserves are used by the parasite, accumulating mosquito resources for the development of oocysts (depicted in dark green) and subsequently its transmissible form, sporozoites (depicted in light green). The ingestion of additional BM benefits the parasite. Parameter sweep of different parasite strength (**c**) and sporogonic cycle $T_{sp}$ (**d**) under different feeding regimes. The colors show the difference in fitness $\Delta F$ of sporozoites (Spz) and eggs in every simulated scenario. The mosquito icons were created with BioRender.com, publication license numbers JK26OOX5C and RU26OOX95.

We further assessed how parasites and vectors compete for resources by varying simultaneously the rate at which parasites scavenge mosquito resources and the number of additional blood meals (Fig. 1c). We then quantified the differences in 'fitness' $\Delta F$ of both parasites and eggs, by calculating the difference in total accumulated energy to the control simulations (i.e. simulations without parasites, and with only one iBM for eggs and parasites, respectively). We observed a strong competitive vector-parasite interaction, with the largest fitness difference detected after three blood meals.

As in this model parasites strongly compete for resources only if the additional blood meal is given during oocyst development, we hypothesized that the duration of the sporogonic cycle $T_{sp}$ will also play a role in the parasite's competition strength. We tested this hypothesis by running different simulations, varying the $T_{sp}$ from 8 to 14 days together with the number of blood meals (Fig. 1d). While all parasites (fast and slow) benefit from two additional blood meals, only parasites with long sporogonic cycles have the opportunity to scavenge resources acquired during subsequent feedings, becoming fitter

than fast parasites. In conclusion, our model of nutrient allocation shows that parasites scavenge progressively more resources from their mosquito host after the second blood meal, suggesting that malaria parasites would benefit from long sporogonic cycles.

## Mosquito metabolism restraints the advantage of shorter sporogonic cycles for malaria transmission

To study whether mosquito metabolism would indeed affect malaria transmission, we developed a stochastic individual-based model that considers female mosquitoes, humans, and parasites. In this model, humans and mosquito individuals are randomly selected during every time step of one day and exposed to specific events with probabilities described in the Methods section. These events include birth, growth, infection, death, and recovery in the case of humans. Our model encompasses the complete mosquito life cycle, including larval and adult stages. Individual larvae exhibit variability in the time of pupation, growth rate, and death rate, which leads to population heterogeneity. We assume that all larval parameters (mentioned above)

depend on ecological variables, such as density, temperature, and carrying capacity (see Methods). After a mosquito reaches the adult stage, it seeks a human blood meal. If successful, fully fed mosquitoes are considered to be digesting for three days before they lay eggs and seek a new blood meal. The cycle of seeking, digesting, and egg-laying is continued during their entire life span, and mosquito death is modeled as an age-dependent function. With these parameters we could model stable mosquito population dynamics (Supplementary Fig. 1a) that matches the age-distribution of mosquitoes observed in field settings[27] (Supplementary Fig. 2).

Infection occurs when a mosquito bites a *Plasmodium*-infected human. To simplify the representation of complex processes involved during parasite infection, we consider three mutually exclusive infection states in mosquitoes (susceptible, exposed, and infectious), and five in humans (susceptible, exposed, infectious, recovered, reservoir). Moreover, parasites are defined only by their sporogonic cycle $T_{sp}$. Thus, upon an infectious bite, mosquitoes carry the parasite (*exposed*), but are *infectious* only after $T_{sp} = 13$ days. For simplicity, we consider that infectious mosquitoes transmit the parasite only to susceptible humans. We implement an incubation period in humans of $T_{IP} = 28$ days, after which exposed humans become *infectious*[28]. During infection, humans have a higher death rate, simulating a high level of parasitemia ($\delta_H = \delta_H + \delta_{p,H}$, where $\delta_H$ is the intrinsic human death rate, and $\delta_{p,H}$ the parasitemia). Only during the infectious state, humans can recover at a rate $p_r$, becoming immune to the parasite (*recovered*). Humans that fail to clear the infection become chronically infected, but harbor lower parasitemia ($0.001\delta_{p,H}$), and a lower probability of transmitting the parasite ($0.1p_{H\rightarrow M}$). We considered these individuals to be asymptomatic carriers, i.e., are in the *reservoir* state. We modeled a homogeneous human population in which individuals die at a fixed rate, and are immediately replaced by new susceptible ones. These infection dynamics, albeit simplified, allowed us to model a natural course of infection, during which more than 60% of individuals remain susceptible (Supplementary Fig. 1b–c and Supplementary Fig. 3).

As a control, we first simulated malaria epidemics excluding the description of resource allocation in the mosquito population (Supplementary Fig. 1a–c). We quantified the exact malaria transmission events by categorizing all mosquitoes in every simulated population into four groups: uninfected, carriers (*exposed* mosquitoes), spreaders (*infectious* mosquitoes that caused one human infection), and super-spreaders (*infectious* mosquitoes responsible for more than one human infection). We observed that spreader and super-spreader mosquitoes are older, and consequently feed multiple times during their life span (Supplementary Fig. 1d–f). However, these mosquitoes are rare in the population (at least one order of magnitude lower than the observed prevalence, Supplementary Fig. 1d), confirming the conventional expectation that *Plasmodium* transmission can be maintained by a few, long-lived vectors that acquire multiple blood meals[29].

To explore the impact of mosquito metabolism on *Plasmodium* transmission, we integrated the effects of nutrient allocation into our individual-based model. For simplicity, we assumed that the parasite's energy accumulated in the sporozoites is related to the mosquito-to-human transmission probability $p_t$. We modified the original model by describing $p_t$ as a function of the number of blood meals acquired during oocyst development (i.e., starting from three days post-infection until the end of the sporogonic cycle $T_{sp}$). Because in our stochastic simulations mosquitoes display individual feeding patterns (i.e., every mosquito will obtain a different number of blood meals during their life span and consequently also during oocyst development), this description of transmission probability ($p_t(N_{BM})$) gives rise to a large heterogeneity in transmission potential. An increase in transmission results in a lower number of eggs per female, allowing us to model the competitive vector-parasite interactions in a simple manner (see Methods). Integrating the properties of mosquito

metabolism (while keeping all other model parameters equal) slightly decreases the infection prevalence (Fig. 2a), a result of the heterogeneity in transmission probabilities and, with it, the number of spreader mosquitoes (Fig. 2b). All other transmission patterns, including the age and blood meal distribution of spreaders, remained similar to the control simulations (Fig. 2c, d).

As the length of the sporogonic cycle is one of the most influential parameters in classical mathematical models of malaria transmission, small reductions in $T_{sp}$ are expected to have a large effect on parasite transmission[26,30]. We next tested how infection dynamics are affected by fast-developing parasites and simulated host populations infected with *Plasmodium* parasites, varying in their sporogonic development from $T_{sp} = 11$ to 13 days. We measured infection prevalence at steady-state in both hosts (at $t = 1,000$ days, Fig. 3a), and the number of spreader mosquitoes (Fig. 3b). As expected from classical predictions, malaria transmission increased with shorter $T_{sp}$ following a linear trend in control simulations. However, when we explicitly modeled within-vector metabolism, this increase was significantly smaller, confirming the results of our resource allocation model showing that mosquito metabolism limits the transmission advantage of short developmental times.

## Mosquito metabolism shapes the evolution of *Plasmodium* parasites

To study whether parasites would indeed benefit from long sporogonic cycles in the context of mosquito metabolism, we next performed evolutionary simulations. First, we allowed mosquito and human individuals to reach stable population dynamics. After this 'burn-in' period of 1000 days, we introduced malaria parasites. We also ensured that infection dynamics reach equilibrium and switched on mutation only after 5000 days. Mutations occur during transmission events and can increase or decrease the length of the sporogonic cycle. We followed the population for further 5000 days and used as control evolutionary simulations without metabolism (Fig. 4a).

There was no difference between the models in the first days of infection dynamics before mutation is switched on. Once evolution started, there was a clear divergence between the simulations excluding (control) and including metabolism (Fig. 4b). In the control simulations, infection prevalence suddenly rose, infecting twice as many humans and mosquitoes. As the fitness of the parasite is independent of mosquito metabolism, a short sporogonic development $T_{sp}$ evolves because fast-developing parasites spread more rapidly in the population, reaching the minimum of $T_{sp} = 5$ days allowed in our simulations (Fig. 4c, purple line). Accordingly, the number of blood meals acquired by every infectious mosquito during the parasite development time shifted during evolution (Fig. 4d, purple bars): most parasites entered their mosquito vectors, and after 5 days, were mature to infect a new human host without acquiring any additional blood meal. In contrast, mosquito metabolism limited the evolution of very short sporogonic development times (Fig. 4c, cyan line). As parasites require blood meals to increase their otherwise low transmission potential, there is selection pressure to acquire at least two blood meals during oocyst development (Fig. 4d, cyan bars), resulting in parasites with an 'optimal' $T_{sp} = 12$ days. Importantly, the evolution of this long sporogonic development time was robust, and independent of the initial conditions: starting the simulations with a low $T_{sp} = 10$ days resulted in the same 'optimal' $T_{sp} = 12$ days (Fig. 4e, f). Thus, our evolutionary simulations show that parasites change their life-history strategies depending on mosquito metabolism, providing a plausible explanation for the long sporogonic cycles observed in natural systems.

## Mosquito longevity and the parasite's scavenging strength determine the evolution of *Plasmodium* sporogonic cycle

In our simulations *Plasmodium* fitness is limited to an optimal $T_{sp} = 12$ days, irrespective of the initial conditions. Increasing $T_{sp}$ beyond

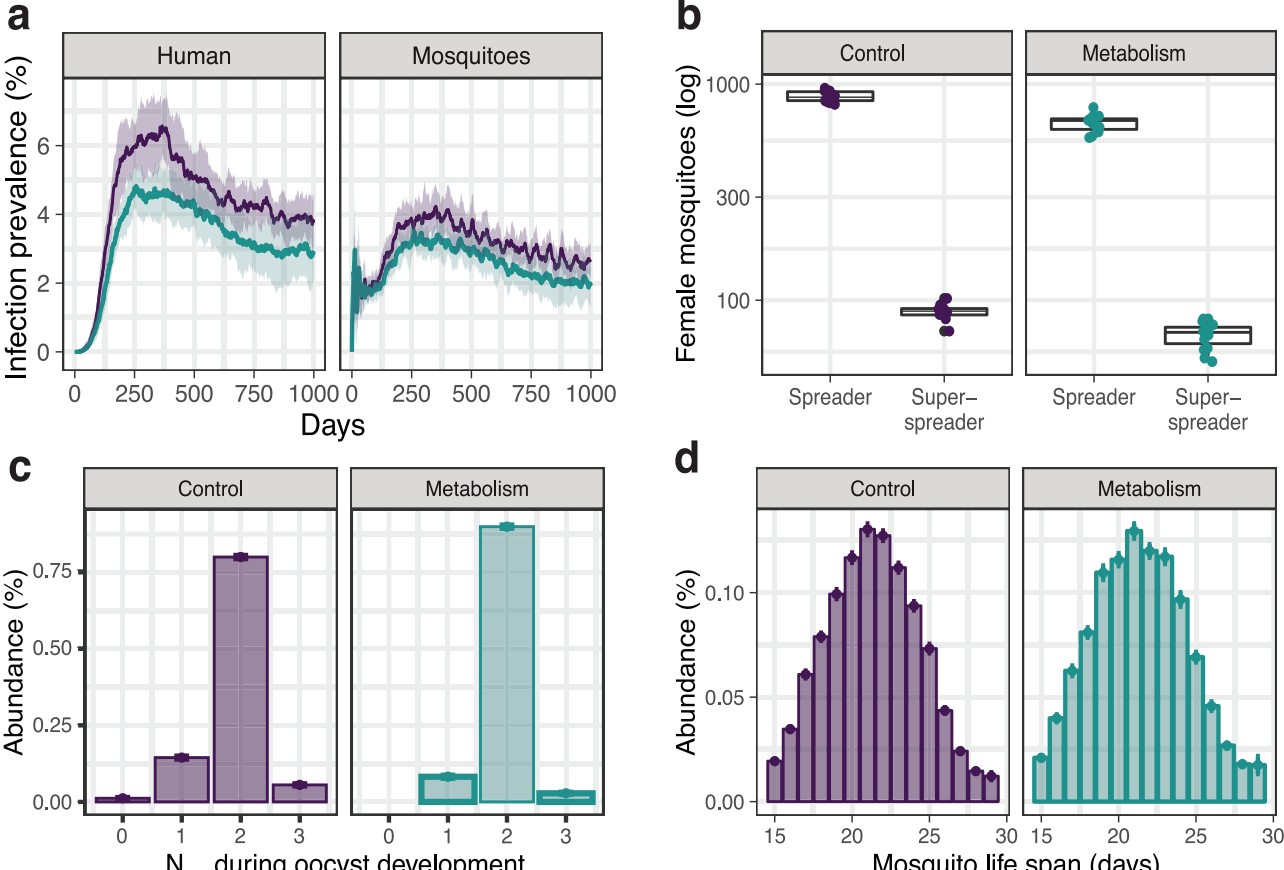

**Fig. 2 | Model of *Plasmodium* transmission including mosquito metabolism.** Summary of $N = 15$ stochastic simulations excluding (control simulations depicted in purple) or including the effects of mosquito metabolism (cyan). **a** Infection prevalence given by the percentage of novel infections in humans and mosquitoes. The solid line depicts the mean and the shaded areas−the standard deviation. **b** Number of spreaders and super-spreaders throughout the *entire* simulation as defined in Supplementary Fig. 1. **c** Distribution of the number of blood meals acquired by single mosquitoes (spreaders and super-spreaders) during oocyst development. **d** Age distribution of spreaders and super-spreaders. Bar plots show the mean, and error bars−the standard deviation. Box plots depict the median with first and third quartile, whiskers depict min and max values.

this value may not be beneficial for the parasite due to the risk of exceeding the mosquito's average lifespan and, consequently, failing to be transmitted. To investigate the influence of mosquito lifespan on *Plasmodium* evolution, we conducted two additional sets of simulations by adjusting the shape parameter $x$ of the Gompertz distribution to model mosquitoes with higher ($x = 0.089$ resulting in median survival of 27 days) and lower ($x = 0.18$ resulting in median survival of 17 days) survival rates compared to our current simulations ($x = 0.169$ resulting in median survival of 19 days, Fig. 5a). Note that modeling a larger decrease in mosquito life span would not assure stable infection dynamics. Therefore, we chose to model only a modest decrease in mosquito life span as a proof of principle. Starting the simulation with a 'fast' $T_{sp} = 10$ days, we observed that malaria parasites evolved an even longer $T_{sp} = 14$ when mosquitoes had an extended lifespan. Conversely, in mosquitoes with a shorter lifespan than in our previous simulations, the evolution of *Plasmodium* was limited already at $T_{sp} = 11.5$, confirming that mosquito lifespan constrains the benefits of an extended sporogony period.

Next, we examined the impact of the parasite's ability to scavenge resources. Since our individual-based model only considers the effects of nutrient allocation, we evaluated the parasite scavenging strength by modifying the slope of the function that describes the transmission probability $p_{M \to H}(N_{BM})$ (Fig. 5b). By changing the relation between number of blood meals and transmission potential, we could investigate how different parasites would evolve. Consistent with our within-vector resource allocation model, we observed that parasites requiring more blood meals to increase their transmission probability ('low scavenging strength') do not benefit as much from long $T_{sp}$ since they would risk exceeding the vector's lifespan. In contrast, parasites with a 'high scavenging strength', i.e., parasites that require fewer blood meals to increase transmission potential ($h = 0.5$), benefit more from an extended $T_{sp}$, evolving a slightly longer 'optimal' $T_{sp} \approx 12$, 2 days. Notably, this effect reaches saturation due to the inherent characteristics of the Hill function, wherein the mosquito-to-human transmission probability $p_{M \to H}(N_{BM})$ approaches 1. Taken together, our simulations show that *Plasmodium*'s evolutionary strategies in response to mosquito metabolism are shaped by the mosquito life span and the parasite's ability to effectively scavenge resources.

## Discussion

We provide a framework that integrates complex mosquito metabolic traits and its interactions with *Plasmodium* parasites into a model of transmission. Our model demonstrates that mosquito metabolism shapes the parasite's evolutionary life-history strategies. Specifically, we show that (1) *Plasmodium* is transmitted by rare long-lived "super-spreader" mosquitoes that take multiple blood meals; (2) successive blood feeding introduces a competitive parasite behavior within the female mosquito that restricts the allocation of nutrients into reproduction, and aids the parasite's own development; and consequently, (3) parasites with long sporogony are selected during evolution.

Our results challenge the current concept that malaria parasites strive to shorten their development to maximize transmission. Given

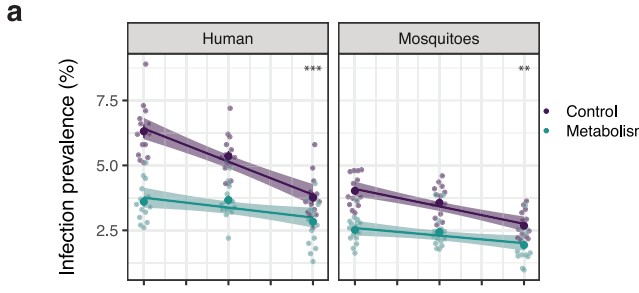

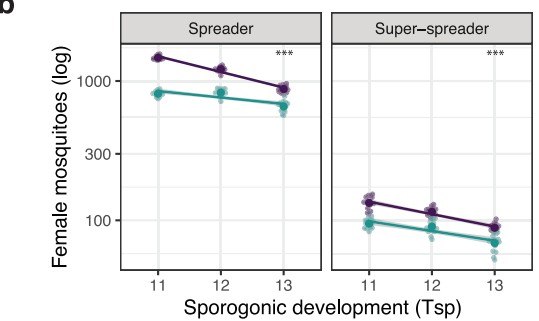

**Fig. 3 | Including the effects of mosquito metabolism restraints the transmission benefits of short sporogonic cycles.** Simulations of malaria transmission assuming different sporogonic cycles ($T_{sp}$). **a** We measure the infection prevalence in both hosts at steady-state, i.e., at $t = 1,000$ days, and **b** the number of spreader and super-spreader mosquitoes in the population throughout the entire simulation time. Note how *Plasmodium* transmission grows with shorter $T_{sp}$ significantly more in the control simulations (purple) than in those including mosquito metabolism (cyan) ($p = 9 \times 10^{-4}$ in humans and $p = 1.6 \times 10^{-3}$ in mosquitoes). Accordingly, the number of spreader ($p = 7.48 \times 10^{-4}$) and super-spreader ($p = 5.8 \times 10^{-4}$) mosquitoes were significantly larger in the control simulations. Difference between groups was calculated with a two sided $t$-test comparing the slopes of linear regressions (lines and shaded areas). ** and *** depict a $p$-value $\leq 0.01$ and $p$-value $\leq 0.001$, respectively. Every dot represents one of $N = 15$ stochastic simulations.

the assumed short life span of the mosquito observed in the field[31,32], it has remained puzzling why malaria parasites develop so slowly. Based on the concept that long sporogony guarantees a large number of sporozoites in the salivary glands, two major explanations have been provided until now. First, a high number of sporozoites is necessary to transmit at least a few to the vertebrate host[33,34]; and second, a high number of sporozoites stimulates mosquito biting to increase transmission[33,35]. Here, we introduce a novel mechanism in which long sporogony is beneficial because parasites scavenge increasingly more resources with successive blood meals. Our model identifies two different mosquito traits that shape the fitness landscape of *Plasmodium* parasites: 1) mosquito lifespan, exerting selection pressure for the parasite to develop fast, and 2) mosquito metabolism which determines the nutrient availability for malaria parasites, resulting in pressure *Plasmodium* to develop slowly (Fig. 6). We, therefore, propose that the long sporogonic cycles observed in nature are not a constraint but rather an adaptation, potentially resulting in a higher transmission success.

A critical parameter in our model is the age-distribution of adult mosquitoes, as it directly determines the number of long-lived vectors and, consequently, of spreaders. The survival rates we use to parameterize our model were obtained under laboratory conditions and result in an older age structure than previously estimated[31,32]. However, the age distribution of our simulated mosquito populations is remarkably similar to wild *An.coluzzi* mosquitoes, as recently measured with novel age-grading methods[27]. Like in our simulations, these natural mosquito populations have a very small proportion of old

individuals (>17 days) that feed frequently (> 4 blood meals). Our work suggests that these rare individuals are the major malaria spreaders and should therefore be studied in more detail.

Our study demonstrates the importance of integrating complex parasite-vector interactions into nested models of *Plasmodium* transmission to realistically predict malaria epidemics and evolution. Since the first mathematical model developed by Ross and MacDonald[29], there has been an expansion of theoretical approaches that consider a variety of geographical, ecological and epidemiological complexities (reviewed in refs. 36–38), as well as multiple mosquito life-history traits (e.g., larval stages[39], biting frequency[40,41], feeding, and movement patterns[42]). However, only very few of these models explicitly consider within-vector parasite development[19,43] beyond the original Ross-MacDonald description[26,30,36,44]. Importantly, parasite evolution has not been traditionally studied in these model extensions, and only one study has investigated the effects of immune dynamics and drug resistance in the human host[45]. Here, we study the mosquito-driven evolution of *Plasmodium* parasites, and show that if mosquito metabolism is explicitly modeled, the exponential relationship between sporogonic cycle and mosquito life-span no longer holds (Fig. 6). Thus, our results indicate that $T_{sp}$ is not as sensitive in determining transmission intensity as was originally suggested[16]. While recent experimental evidence has shown that reductions in sporogony time does not diminish the infectivity to primary hepatocytes[14], these early sporozoites are very low in numbers. Whether such low sporozoite numbers will be sufficient for a successful infectious bite, remains to be demonstrated. Further studies are needed to assess the effect of shorter EIP on parasite transmission efficiency.

By conceptualizing complex metabolic processes within the mosquito, we have obtained a new perspective for understanding malaria transmission and evolution. This work raises new questions that need to be addressed both experimentally and theoretically. Experimentally, the relationship between parasite competition strength and the number of blood meals must be quantified. How strong will mosquito reproduction be impaired by malaria parasites during natural feeding regimes (more than two blood meals)? Moreover, we assumed that the metabolic energy accumulated by the parasite can be translated into transmission probability. Our simulations show that the relationship between the number of blood meals and transmission is essential in determining *Plasmodium*'s evolutionary outcomes, calling for more empirical work in quantifying this relationship: how do multiple feedings affect the number, and/or, quality of sporozoites, and their transmission potential into a vertebrate host? Recent studies have shown that additional blood meals influence parasite growth, resulting in larger oocysts[13,46], and an accelerated invasion of sporozoites of the mosquito salivary glands reducing the time potentially required for transmission[16]. Our work suggests that the fitness of those early sporozoites should be low given the reduced metabolic resources they acquired during their fast development. Do parasites with different $T_{sp}$ display differences in transmission potential? Measuring sporozoite numbers, and quality in infected mosquitoes fed under different regimes is necessary to further understand the effect of mosquito metabolic resources on *Plasmodium* development. Additionally, given that malaria parasites exhibit adaptation to their vertebrate hosts[2,6,10], it seems plausible that they also show phenotypic plasticity depending on the metabolic status of their mosquito-host: would they sense and modulate their sporogonic cycle in response to the nutrients available in different mosquitoes?

Theoretically, extensions of the model will help to test hypotheses that are difficult to study in an experimental or field setting. For example, shifts in vector feeding schedules and life-spans can be studied computationally to assess the effect of different mosquito species on parasite transmission. Additionally, several genetic and environmental factors have also been identified as determinants in *Plasmodium* sporogony[26,47], including the mean environmental temperature[48–50], genetic diversity of both the vector and the parasite[51],

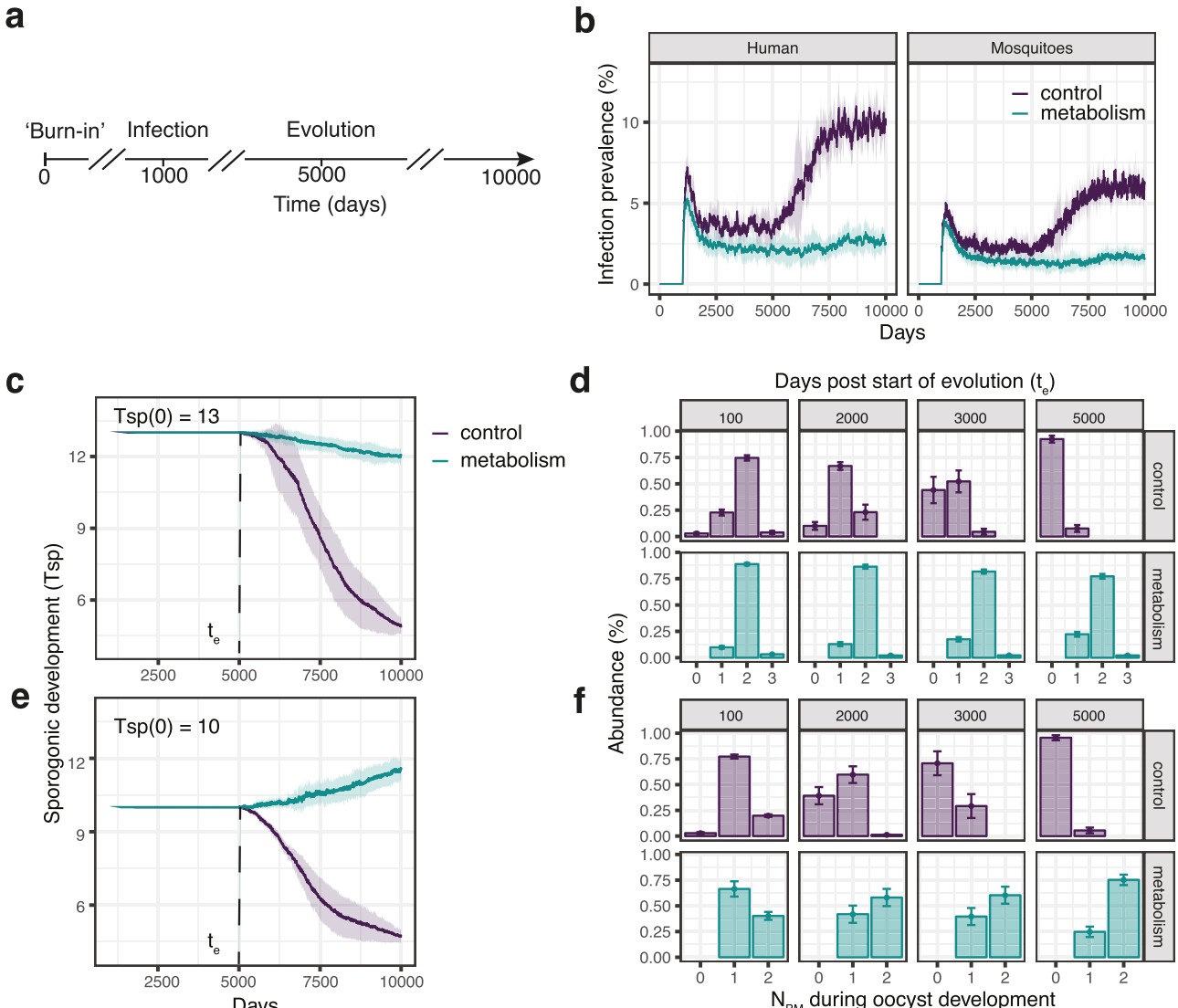

**Fig. 4 | Mosquito metabolism shapes *Plasmodium* evolution. a** Simulation protocol for evolutionary simulations. After 1000 days of `burn-in' period, we infect 20% of the human population with malaria parasites. We allow infection dynamics to reach a steady state until 5000 days, where we switch on mutation. Simulations end after 10,000 days. We compare the evolutionary patterns observed in our original model (control, purple) to those emerging in the model including the effects of metabolism (cyan). **b** Time course of infection prevalence. Note how infection increases after mutation is turned on only in the control simulations. **c, e** Time course of the sporogonic development $T_{sp}$. **d, f** Distribution of blood meals acquired by individual mosquitoes during oocyst development. Every

column summarizes the blood meals of individual mosquitoes 100, 2000, 3000, and 5000 days after turning on evolution, in control (upper row) and simulations including vector metabolism (lower row). To test the effect of initial conditions we run two sets of simulations: one with $T_{sp}(0) = 13$ (**c, d**) and another with $T_{sp}(0) = 10$ (**e, f**). Independent of the initial conditions, a minimal $T_{sp} = 5$ naturally evolves in the control simulations, while mosquito metabolism maintains a long $T_{sp} \approx 12$. Solid lines in the time courses depict the mean, with shaded areas displaying standard deviation. Bar plots show the mean, and error bars the standard deviation. Summary of $N = 10$ stochastic simulations.

as well as nutritional status during larval stages[47,52,53]. How these processes would interact together with the metabolic parasite-vector competition presented here can be addressed in future extensions of our model. Thus, our theoretical approach sets the basis for future investigations on the vector-parasite pairs that favor transmission, possibly revealing unexpected strategies parasites naturally evolve.

## Methods

### ODE model of resource allocation

We model the mobilization of nutrients after a blood meal with a system of ordinary differential equations that consider three vector-specific compartments (blood digested during a blood meal ($B$), within-host energy reserves ($R$), and reproductive energy for developing eggs ($E$)), and two parasite-specific compartments (internal energy resources for oocysts ($O$) and developing sporozoites ($S$))

(Fig. 1). The energy flow from one compartment to the other is activated with rectangular pulse wave functions, described by:

$$\sigma(t) = \begin{cases} 0, & \text{if } t < \tau \ \& \ t - \tau - n_i T_\sigma > \lambda \\ 1, & \text{otherwise} \end{cases} \quad (1)$$

Here, a pulse starts after a delay of $\tau$ and lasts for a duration of $\lambda$. The pulse is repeated with a period $T_\sigma$ and will be repeated for $N_{BM}$ blood meals, thus $\sum_1^{N_{BM}} n_i T_\sigma$. For every process different values of $\tau$, and $\lambda$ were chosen as described below. After the ingestion of a blood meal $\sigma_{BM}(t)$, the blood can be either digested at a rate $\delta_B$, or its metabolic energy can be mobilized to the mosquito reserves $R$ after a period $\tau_{BM} = 24$ h. This blood-meal activates the signal for the steroid hormone 20E synthesis in the ovaries $\beta_E(t)$. A proportion of the host resources is then mobilized to the ovaries and is accumulated by

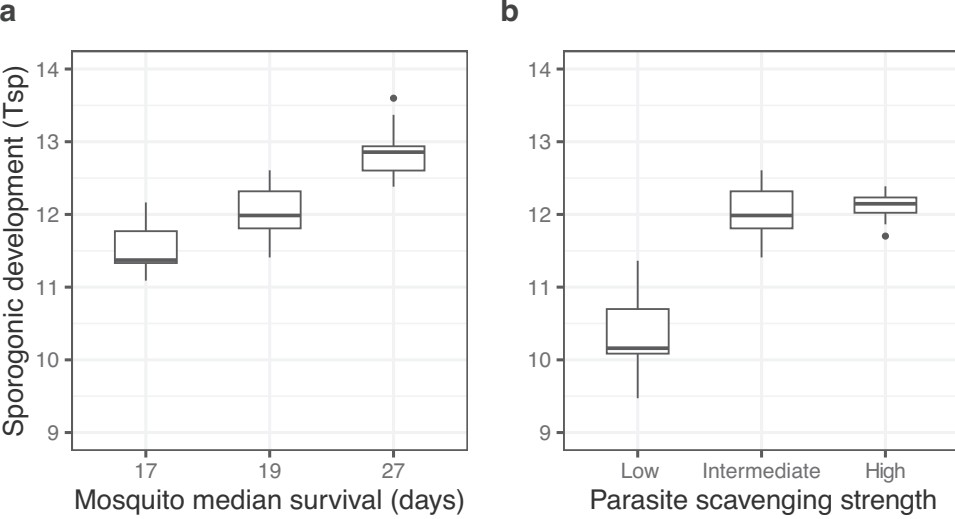

**Fig. 5 | Mosquito longevity and parasite's strength determine the evolution of *Plasmodium* sporogonic cycle.** Evolved sporogonic development time ($T_{sp}$) at the end of the evolutionary simulations (at $t = 10{,}000$ days). **a** We simulated two additional mosquito populations with different age-dependent death rates by changing the shape parameter ($x$) of the Gompertz distribution, resulting in younger ($x = 0.18$) and older ($x = 0.089$) mosquitoes compared to our previous simulations. **b** We also modified the slope of the Hill function describing the mosquito-to-human transmission probability $p_{M \to H}(N_{BM}) = p_0 + \frac{n_{BM}}{n_{BM} + h}$, effectively modeling parasites with low ($h = 2$), intermediate ($h = 1$), or high ($h = 0.5$) scavenging strength. Note that the simulations ran with parameters $x = 0.16$ and $h = 1$ correspond to the results depicted in Fig. 4. All simulations were started with a fast initial sporogonic development ($T_{sp}(0) = 10$ days). Box plots show the median with first and third quartile, whiskers depict min and max values. Summary of $N = 10$ stochastic simulations.

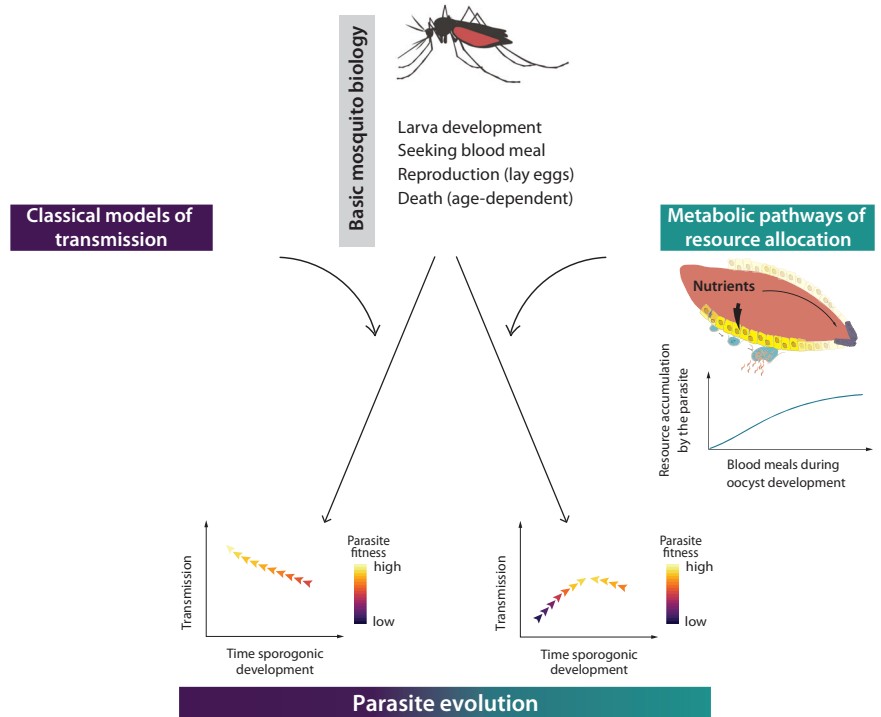

**Fig. 6 | Evolutionary modeling predicts that mosquito metabolism shapes life-history traits of *Plasmodium* parasites.** By only considering basic life-history traits of mosquitoes (larval stages, biting behavior, reproduction, and death) and a simple within-vector *Plasmodium* development (an exponential relationship between mosquito longevity and transmission probability), classical models of malaria transmission predict that a reduction of the sporogony time ($T_{sp}$) increases parasite's fitness and aids transmission success (left). In contrast, when mosquito metabolism (and with it the associated parasite competition for mosquito resources during multiple blood meals) is also included, our model reveals a novel evolutionary scenario in which intermediate long sporogony is optimal for transmission (right). The mosquito icon was created with BioRender.com, publication license number RU26OOX95.

developing eggs ($E$). The remaining resources are invested into other physiological processes, including immunity, physical activity, and waste, at a rate $\delta_R$. If a blood meal is infected with *Plasmodium*, oocysts ($O$) grow by accumulating resources taken from the vector's reserves at a constant rate $\beta_P$. Once mature, oocysts transfer their internal energy into sporozoites ($Sp$) at a rate $\gamma(t)$. The dynamics of $\beta_E(t)$ are explicitly modeled by Eq. (7), where $a$ and $b$ represent the activation and inhibition rate of reproductive investment, respectively. Periodic

blood meals, the thereby induced resource allocation, and the blood meal dependent signal for 20E activation occur with a period $T_\sigma = 7$ days, durations $\lambda_{BM} = 6$ h, $\lambda_R = 60$ h, and $\lambda_{\beta_E} = 24$ h, and delays of $\tau_{BM} = 6$ h, $\tau_R = 30$ h, and $\tau_{\beta_E} = 6$ h, respectively. Similarly, the time for oviposition is activated with a time delay $\tau_{d_E} = 60$ h and lasts $\lambda_E = 6$ h. The model is given as follows:

$$\dot{B} = \sigma_{BM}(t)(1 - B) - \sigma_R(t)BR(1 - R) - \delta_B B \tag{2}$$

$$\dot{R} = \sigma_R(t)BR(1 - R) - c\beta_E(t)RE(1 - E) - \beta_P RO(1 - O) - \delta_R R \tag{3}$$

$$\dot{E} = c\beta_E(t)RE(1 - E) - \sigma_E(t)\delta_E E \tag{4}$$

$$\dot{O} = \beta_P RO(1 - O) - \gamma(t)O \tag{5}$$

$$\dot{S} = \gamma(t)O \tag{6}$$

$$\dot{\beta}_E = \sigma_{\beta_E}(t)a(1 - \beta_E) - b\beta_E \tag{7}$$

$$\gamma(t) = \begin{cases} 0, & \text{if } t \le T_{sp} \\ \gamma, & \text{otherwise}. \end{cases} \tag{8}$$

Note that this is a conceptual framework that models the metabolic processes in the mosquito. Importantly, not all of the processes described here can be measured experimentally. Therefore, most parameter values have been chosen to qualitatively match empirical data describing the temporal dynamics of 20E activation and oviposition[18], as well as infection dynamics typically observed during *Plasmodium* infections. All parameters are fully described in Table 1.

**Parameter sweeps.** To test for the effect of different parameter values, we performed several parameter sweeps, varying one parameter at a time in small intervals. We studied the effects of different parasite strengths in allocating resources by changing $\beta_P$ from [0.0-0.1] in 0.05 steps. The impact of the number of blood meals was tested by running several simulations altering $N_{BM}$ from [1-5] days. Finally, we simulated five different sporogony times $T_{sp}$ ranging from [11 - 15] days in 2 days intervals.

**Fitness calculations.** We define arbitrary fitness functions for eggs and parasites. Here, reproductive fitness is defined as the sum of the maximal energy accumulated during every blood meal (Eq. (9)), where as parasite fitness is defined the maximal energy accumulated in the sporozoites compartment (Eq. (10)). The differences in fitness $\Delta F(E)$ (Eq. (11)) and $\Delta F(S)$ (Eq. (12)) are computed by subtracting the control scenarios (i.e., simulations without parasite and with only one infected blood meal for eggs and parasites, respectively) from the simulation of interest.

$$F(E) = \sum_{i=0}^{N-1} \max(E(\tau_i)); \text{ with } iT_\sigma \le \tau_i < (i+1)T_\sigma \tag{9}$$

$$F(S) = \max(S(t)) \tag{10}$$

$$\Delta F(E) = F(E)_{iBM} - F(E)_{BM} \tag{11}$$

$$\Delta F(S) = F(S)_{N > 1} - F(S)_{N = 1} \tag{12}$$

**Table 1 | Model parameters of the resource allocation model**

| Parameter | Value [Unit] |
|---|---|
| **Blood meal parameters** | |
| Feeding interval $T_\sigma$ | 7 [days] |
| Duration of feeding $\lambda_{BM}$ | 6.00 [h] |
| Time delay of feeding $\tau_{BM}$ | 6.00 [h] |
| Duration of 20E activation $\lambda_{\beta_E}$ | 24.00 [h] |
| Rate of 20E activation $a$ | 0.5 [h$^{-1}$] |
| Rate of 20E inhibition $b$ | 0.5 [h$^{-1}$] |
| **Host parameters** | |
| Duration of nutrient transport into the reserve compartment $\lambda_R$ | 60.00 [h] |
| Rate at which energy flows into physiological processes $\delta_R$ | [$5 \times 10^{-4}$ h$^{-1}$] |
| Energy flow rate for egg development $c$ | 0.3 [h$^{-1}$] |
| Degradation rate of reproductive energy $\delta_E$ | 0.5 [h$^{-1}$] |
| Duration of oviposition $\lambda_E$ | 6.00 [h] |
| Time delay of oviposition $\tau_{d_E}$ | 66 [h] |
| **Parasite parameters** | |
| Energy flow rate for parasite development $\beta_P$ | 0.045 [h$^{-1}$] |
| Rate of sporozoite development $\gamma$ | 0.1 [h$^{-1}$] |
| Sporogony time $T_{sp}$ | 11 [days] |
| **Initial conditions** | |
| Blood meal $B(0)$ | 0 [AU] |
| Internal reserves $R(0)$ | 0.8 [AU] |
| 20E activation $\beta_E(0)$ | 0 [AU] |
| Reproductive energy $E(0)$ | 0.1 [AU] |
| Energy of oocysts $O(0)$ | 0.01 [AU] |
| Energy of sporozoites $S(0)$ | 0 [AU] |

## Individual-based model of malaria transmission

We develop an individual-based model consisting of three actors (humans, female mosquitoes and parasites) and four types of events (birth, growth, infection, recovery -in humans-, transmission, and death). The basic time step of the model is one day, during which every human host and vector is chosen in a random order and confronted to one of the randomly chosen events. During infection, parasites are embedded in every host, and will be copied into a new host upon every transmission event. Humans and vectors age over time, i.e., their age is updated after each time step. The cycle is repeated over several vector generations. All parameters are fully described in Table 2. The following is a detailed description of individuals and events.

**Mosquitoes.** We consider both phases of the mosquito life cycle and explicitly model larvae and adults. Individual larvae die at a temperature dependent rate, described as a quadratic function: $\delta_L(T) = a_\delta T^2 - h_\delta^2 + k_\delta$. Larvae develop into adults only after a pupation time that follows an exponential function: $T_P(T) = \frac{m_t}{1 + \exp(Th_t)} + k_t$. After this pupation time, we allow larvae to grow with a density-dependent rate $\gamma(L) = 2^{-L/K_L} m_\gamma$, where $K_L$ is the carrying capacity of the assumed aquatic habitat, and $m_\gamma$ the maximal daily growth rate. We consider individual effects by assuming that every parameter $j$ consists of population and individual effects, as follows: $\theta_{ij} = \mu_j e^{\eta_{ij}}$. Here $\mu_j$ is the population parameter, and $\eta_{ij} \sim \mathcal{N}(0, \omega_j^2)$ is the random effect for every individual larvae $i$. The larval parameters given in Table 2 were obtained by fitting a mechanistic mathematical model to data of larva development at 28°C[54]. Of note, all rates ($r$) were converted into their respective probability as described by $p = 1 - \exp(-rt)$.

We allow female mosquitoes to seek a blood meal immediately after emerging as adults, as previously modeled[55]. During their seeking state, they bite a randomly chosen human depending on the number of

## Table 2 | Model parameters of the individual-based model

| Parameter | Value [Unit] |
|---|---|
| Time step | 1 [day] |
| Environment temperature $T$ | 28 [˚C] |
| **Host parameters** | |
| Carrying capacity $K_H$ | 1000 individuals |
| Death rate $\delta_H$ | 1 [year⁻1] |
| **Vector parameters** | |
| Carrying capacity of aquatic habitat $K_L$ | 5540 individuals |
| Coefficients of larval death rate $\delta_L(T) = aT^2 - bT + c$ | |
| $a_{pop}$; $\eta_a$ | 0.00288; 1.12 |
| $b_{pop}$; $\eta_b$ | 0.00663; 0.645 |
| $c_{pop}$; $\eta_c$ | 0.0381; 0.701 |
| Coefficients of larval growth rate $\gamma(L) = 2^{-L/K_L} m_\gamma$ | |
| $m_{\gamma,pop}$; $\eta_{m_\gamma}$ | 1.19; 0.513 |
| Coefficients of pupation time $T_P(T) = \frac{m_t}{1 + \exp(Th_t)} + k_t$ | |
| $m_{t,pop}$; $\eta_{m_t}$ | 9.63; 0.104 |
| $h_{t,pop}$; $\eta_{h_t}$ | 0.0996; 0.0258 |
| $k_{t,pop}$ : $\eta_{k_t}$ | 2.95; 0.188 |
| Mosquito birth probability | 0.03[55] |
| Number of eggs per oviposition | $\mathcal{N}(\mu = 55, \sigma = 24.5)$ |
| Mosquito biting probability | 0.97[55] |
| Mosquito hazard/death rate $h(age) = b \exp(x\,age)$ | $x = 0.16$, $b = 0.007$ |
| **Parasite parameters** | |
| Parasite incubation period in humans $T_{IP}$ | $\mathcal{N}(\mu = 28, \sigma = 0.25)$[28] [day]⁻¹ |
| Parasite incubation period in mosquitoes $T_{sp}$ | $\mathcal{N}(\mu = 13, \sigma = 0.25)$ [day]⁻¹ |
| Parasitemia in humans $\delta_{p,H}$ | 0.001 |
| **Infection parameters** | |
| Transmission probability mosquito-to-human $p_{M \to H}$ | 0.7 |
| Transmission probability human-to-mosquito $p_{H \to M}$ | 0.7 |
| Recovery probability in humans $p_r$ | 0.75 |
| **Initial conditions** | |
| Larva initial population $L_O$ | 100 |
| Mosquito initial population $M_O$ | 100 |
| Human initial population $H_O$ | 1000 |

human hosts in the population, as follows: $p_b' = p_b\left(\frac{N_H^3}{N_H^3 + h^3}\right)$, where $p_b$ is the maximal biting probability, $N_H$ is the number of human hosts, and $h$ is the human population at which the probability is half maximal. After a successful blood meal, every mosquito digests for a period of approximately three days, after which it develops eggs. The number of eggs developed per female mosquito $N_E$ follows a normal distribution $\mathcal{N}(\mu_E, \sigma_E)$, as typically observed in our laboratory. We assume that not all eggs survive, and model the effective probability of birth as a density-dependent function $b = \hat{b}(1 - L/K_L)$, where $\hat{b}$ is the maximal birth probability, $L$ is the number of existing larvae, and $K_L$ is the carrying capacity of the aquatic habitat. All surviving eggs contribute immediately to the larvae population.

Mosquitoes die with an age-dependent death rate described with a Gompertz distribution with shape $x = 0.16$ and rate $b = 0.007$. This death rate function was obtained by fitting several distributions to survival curves of mosquitoes reared at 28˚C in our laboratory. Note that in our laboratory mosquitoes did not show different survival profiles when infected, and therefore we do not simulate any infection-related survival cost.

**Humans.** We consider a homogeneous human population, in which every human has a death rate $\delta = \delta_H + \delta_{p,H}$, where $\delta_H = 1$[year⁻¹] is the intrinsic death rate, and $\delta_{p,H} = 1 \times 10^{-3}$ is the increase caused by the

parasitic infection (see below). Humans live on average one year. While this is short, we do this to assure that sufficient susceptible individuals are available and new infections can constantly occur. For simplicity, we keep the human population constant, i.e., when an individual dies due to background mortality, or infection, it will be immediately replaced by a new susceptible individual.

**Parasite.** In our model, *Plasmodium* parasites are only described by the duration of their sporogonic development $T_{sp} \sim \mathcal{N}(\mu = 13, \sigma = 0.25)$, thus generating a heterogeneous parasite population. Parasites mutate upon every transmission event with a probability $p_m = 0.5$ by randomly increasing or decreasing their $T_{sp}$ by 0.5. We set an arbitrary minimal value of $T_{sp} = 5$ days.

**Infection dynamics.** For simplicity, we do not model the individual stages of parasite development within each host. Instead, we capture infection dynamics with a classical "Susceptible-Exposed-Infectious-Recovered-Susceptible" (SEIRS) model for humans, and an "Susceptible-Exposed-Infectious" (SEI) model for mosquitoes. Susceptible mosquitoes are exposed to the parasite with a probabilty $p_{H \to M}$ when feeding on an infectious human. Mosquitoes become infectious after the parasite has completed its development, i.e., after a sporogonic development time $T_{sp}$. Infectious vectors can transmit the parasite to susceptible humans upon their next blood meal with a transmission probability $p_{M \to H}$. Like in mosquitoes, exposed humans become infectious after a development period $T_{IP}$. During this period, their death rate increases by $\delta_{p,H}$, thereby modeling parasitemia. Infected individuals recover from the infection with a probability $p_r$. Humans that fail to recover become chronically infected with the parasite, and show lower transmission probability ($0.1p_{H \to M}$) and parasitemia ($0.001\delta_{p,H}$), effectively modeling asymptomatic carriers.

### Mosquito metabolism
For simplicity, we include mosquito metabolism into the iBM by assuming that the energy accumulated into the sporozoites is translated to the mosquito-to-human transmission probability $p_{M \to H}$. Effectively, every parasite has an intrinsic basic probability of transmission ($p_0 = 0.2$) which grows after the acquisition of blood meals during oocyst development (i.e., three days after ingestion until $T_{sp}$) as described with a simple Hill function $p_{M \to H} = p_0 + \frac{n_{BM}}{n_{BM} + 1}$. Note that with this description and our chosen parameters, a parasite requires $n_{BM} = 1$ to obtain the same transmission probability as in our control iBM. Similarly, the number of eggs developed per female will be dependent on the parasite 'strength', given by the same $p_{M \to H}$, as follows: $\hat{N}_E = 0.85 N_E - 0.4 p_{M \to H}$, where $N_E$ is the number of eggs per batch, per individual female as described above. All other agents, events, and parameters are the same as in the model excluding metabolism (which we refer throughout this manuscript as control).

**Model initialization.** The model was initialized with 100 larvae and 100 adult females, each with a random age between 1–10 days. Similarly, 1000 human hosts were set at the start of the simulations. 20% of the human population are initialized as asymptomatic carriers.

**Life span calculations.** We categorized mosquitoes in every simulated population into four groups: uninfected, carriers (*exposed* mosquitoes that died before becoming infectious), spreaders (*infectious* mosquitoes that caused one human infection), and super-spreaders (*infectious* mosquitoes responsible for more than one human infection). We compute the life span by recording the age (in days) at which mosquitoes died, and calculated the median life span per category, per simulation.

### Model assumptions
The purpose of our theoretical framework is to explore the role of within-mosquito physiology on the evolutionary strategies of malaria

parasites. Therefore, our models are necessarily an abstraction of complex metabolic and transmission processes. Wherever possible, parameters were obtained by quantifying data generated in our laboratory, or adopted from the literature. In the individual-based model, we prioritized parameter choices that resulted in realistic infection dynamics. Therefore, the modeled human population is homogeneous, and remains constant throughout the simulations. Humans are assumed to have a short life span of one year. While unrealistic, we do this to prevent the infection from saturating, thereby ensuring that sufficient susceptible individuals are available. For simplicity, we model only one mosquito species, ignore climate seasonality, and do not consider super-infection in either of the hosts. This simplification allows us to focus on the role of mosquito metabolism on malaria transmission and evolution.

The ODE model of resource allocation simulates the energy flows induced by metabolic changes. Because there is currently no data available of these dynamic processes, parameter values were chosen to match the dynamics typically observed in the activation of 20E after a blood meal[18], and during parasite development. We assume that parasites scavenge resources at a constant rate $\beta_P$ proportionally to their resources. As a result, less energy will be scavenged by the early-stage oocysts compared to the late-stage ones. We also assume that the energy scavenged by developing oocysts will be completely transferred to sporozoites. While there is some experimental evidence that the relationship between oocyst and sporozoite numbers saturates at large parasite densities[56], we do this to keep the model simple. Importantly, we do not model parasite numbers, densities, or size, but only metabolic energy. A high accumulation of energy could be interpreted as either multiple small occysts, or few large ones. Similarly, a high amount of energy accumulated by sporozoites could translate into large numbers, and/or quality. Accordingly, we intuitively assume that the metabolic energy determines transmission probability in the evolutionary individual-based model. Future experimental studies are imperative to determine how the metabolic energy accumulated by oocysts would affect transmission potential, and validate our model implications.

### Reporting summary

Further information on research design is available in the Nature Portfolio Reporting Summary linked to this article.

## Data availability

The data generated in this study have been deposited in the Edmond database under the link: https://doi.org/10.17617/3.41DUIB.

## Code availability

The simulation framework for the study was developed in C++ and incorporates the Boost library (version 16). For the analysis of the simulated data, R (version 3.5) was utilized with the following packages: plyr, data.table, dplyr, tidyverse for data processing and ggplot2, cowplot for data visualization. All relevant code for the model as well as the accompanying scripts can be accessed under: https://gitlab.mpcdf.mpg.de/vectorbiology/mosquito_metabolism/resourceallocation_release. The code for the ODE model of resource allocation was developed in R (version 3.5) with the additional packages: deSolve for solving ordinary differential equations, reshape and viridis for data processing and visualization, respectively. The scripts can be found at: https://gitlab.mpcdf.mpg.de/vectorbiology/mosquito_metabolism/vectorbornemodel_release.

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

## Acknowledgements

We thank Enrico Sandro Colizzi, and Bram van Dijk for comments on earlier drafts of this manuscript. We are also very grateful to Sarah Reece, Nicole Mideo, Lauren Childs, Megan Greischar, Angelo Valleriani, and all members of the Vector Biology Unit for fruitful discussions and helpful comments.

## Author contributions

PCB, GC, LL, and EAL conceived the study and conceptually designed the model framework. PCB performed the computational modeling, programming, data analysis, and visualization. PCB, GC, LL, and EAL wrote the manuscript.

## Funding

## Competing interests

The authors declare no competing interests.
