## [Peer Review File · Nature Communications]

Evolutionary modelling indicates that mosquito metabolism shapes life-history strategies of *Plasmodium* parasitesREVIEWER COMMENTS

Reviewer #1 (Remarks to the Author):

In this paper, the authors show how slow parasite development can benefit *Plasmodium* parasites in the mosquito host by allowing it to take advantage of the extra resources to developing oocysts provided by the host taking multiple blood meals, and that the balance between within-host selection for slow development and between-host selection for fast development can explain the observed duration of the sporogonic cycle. They do this by deriving a simplified, but mechanistic, model for both mosquito reproduction and *Plasmodium* development, and then nesting this model in an individual-based model of *Plasmodium* transmission among human and mosquito hosts. I thought this paper was really interesting, if a bit difficult to read at times. I am hoping my comments below will help to clarify my points of confusion and help highlight the key results that make this paper fascinating.

My major structural comment is that a significant improvement in this paper would result, in my opinion, from rearranging the model presentation. In particular, I think you should start with the model of oocyst development (Fig. 2). That allows you to move forward in your manuscript your really interesting and absolutely key result (Fig. 2D) that parasites can benefit from a long sporogonic cycle because it allows them to take advantage of multiple blood meals to increase their production of oocysts before switching to sporozoite production. I would move all of the results shown in Fig. 1 into the Supplemental Material because the results shown there are pretty intuitive. I think you can just say something like, "Although our within-vector model shows that parasites benefit from a long sporogonic cycle, in nature that is a risk, as hosts must live long enough to take multiple blood meals. Given that mosquito mortality is age-dependent, the mosquitoes face an important trade-off between a long sporogonic cycle maximizing within-vector fitness and it potentially jeopardizing transmission of sporozoites to the human intermediate host. In the Supplemental Material we use an individual-based model of transmission with a fixed sporogonic cycle to show that although most mosquitoes do get multiple blood meals, very few live long enough to take multiple blood meals after an infectious blood meal, and these are very important to overall transmission." Then, in the Supplemental Material, I would replace Fig. 1F with a figure that shows the average number of blood meals following an infectious blood meal to emphasize how rare it is to obtain multiple blood meals following the infectious one (as you do in Fig. 3C). This allows you to really simplify the model presentation because rather than explaining the individual-based model, explaining the within-host model, and then explaining how to embed the within-host model into the individual-based model, you can just do the last two.

I also felt that the Discussion section did not do enough to place this new model within the broader modeling literature. Although malaria is not my area of research, I am familiar with the evolutionary epidemiology literature – the approach you are using here of combining within-host and between-host models is often called a "nested model" approach. A quick Google Scholar search for ("nested model" + *Plasmodium*) brings up 408 results, including some that you cite (such as Costa et al. 2018 Nature Communications, although that was not actually cited in the Discussion section) and many that you do not. No doubt most of those models are not relevant here, but including something about how your

nested model differs from other nested models that have been used to study malaria evolution would be useful. Perhaps some of this is on lines 197-204, but it is unclear how those models were constructed, and I know that there are a lot of within-host models that do include relatively complex descriptions of the Plasmodium life cycle (e.g., Mideo et al. 2008 American Naturalist) – have any of these been nested into a between-host model?

Specific comments:

Line 38-39: This is a really fascinating statement, and one I would like to see explained a bit more. When you say that the, “parasites rely on the same resources,” do you mean that very specifically, that they use essential yolk proteins for oocyst and sporozoite development? Also, HOW does reproductive investment and nutrient allocation influence oocyst and sporozoite development? Is that the more the mosquito invests in reproduction, the more parasite reproduction occurs? The next sentence suggests that it is only surplus metabolic resources that are available to the parasite, so it would seem possible that the amount of surplus is relatively unaffected by the total allocation to reproduction (e.g., if it takes 1 unit of resource to make an egg, and the mosquito allocates 4.5 units of resource to reproduction, then the mosquito will make 4 eggs with 0.5 units surplus; if it allocates 8.5 units of resource, then it makes 8 eggs and still has only 0.5 units of surplus). I would love to have a bit more explanation of what is going on here, and it seems like this process is quite critical to understanding the rest of the manuscript.

Line 43-44: Can you explain what you mean by, “How would malaria parasites adapt to the metabolic states of their mosquito host?” What is the evolving trait in this adaptation? Is it the relative allocation between growth and reproduction, as alluded to in the second paragraph?

Lines 60-83: I really struggled to understand the structure of the model. I realize that Nature Communications puts the methods at the end, but there needs to be enough explanation of the model in the Results to understand how the Results were derived, and that was not the case for me. For instance, you say that (line 61-62) that randomly chosen humans and mosquitos experience either birth, death, or infection. How do you decide which event occurs? Then the very next sentence says that mosquitos also grow and pupate – how does that fit into the “birth, infection, and death” of the previous sentence? How is growth, death, and pupation affected by ecological variables? Next, you go on to describe the developmental cycle of the Plasmodium, but you initially just encapsulate the entirety of mosquito development into the delay between exposure and infectiousness. Next, you include the process of recovery for the human population, which again does not fit into your statement that the stochastic processes are “birth, infection, and death.” Then you say that individuals that fail to recover become chronically infected, but with lower parasitemia (but not a lower probability of death, contradicting a previous statement that you assume infection increases probability of death because of a high level of parasitemia). But if recovery is a stochastic process, then any individual, at any time, could recover – how then do you decide when an individual has become chronically infected?

Line 63: Is this saying the traits are (1) the time of pupation; (2) growth rate; (3) death rate? Or are you saying the traits are the rates of pupation, growth, and death?

Line 77: What does the “EIP” in T_EIP stand for?

Fig. 1D: There is very little variation around the percentages, but if you add the percent of female mosquitoes in each category up, they look to be over 100% - the figure indicates that very close to 100% of females are uninfected. Maybe this is because the results are presented on a log scale – you might actually put in the caption what the average percent of females in each category is to avoid confusion.

Line 90-92: Why is this surprising? This seems completely intuitive since mosquitoes have to live long enough to become infected (which may be rare), then live long enough to become infectious (which takes at least 13 days), then live long enough to take a blood meal (and become a spreader), then live long enough to take a second blood meal (at least three more days to become a super-spreader).

Lines 96-116: Based on the structure of the within-host model of energy allocation in Fig. 2A, I do not see how it is possible for Plasmodium oocysts (O) to not compete with mosquito oocysts (E) for reserves (R) even after the first blood meal because (line 105) they take, “resources from the vector’s reserves at a constant rate β_P .” Without an explicit delay, the Plasmodium will be constantly taking some energy from reserves; I assume this is happening, just at a very low rate compared the rate of allocation to eggs, and that the total rate of energy consumption from reserves depends on the number of Plasmodium oocysts. That would make sense to me: after a single blood meal, there are a very low number of oocysts, so even though they are taking some reserves away from possible egg production, you don’t see the impact; but when there are multiple blood meals, by the second blood meal O is large enough for the energy drain from reserves to impact mosquito reproduction. If that is what is happening, I would say that it is not that (line 112), “the parasite is not in direct competition with its vector.” It is always in direct competition with its vector, but you can only see that competition when O is large enough to have an impact, which only occurs if there are multiple blood meals. To put it another way, if you assumed that the single infectious blood meal contained a very large number of developing oocysts, so $O(0)$ was appreciably large, I hypothesize that you would see an impact on mosquito reproduction.

Line 142-143: How do classical models incorporate T_{sp} ? I realize this is explained in detail in the Discussion, but a brief explanation along the lines of the caption of Fig. 6 might answer some questions for a reader here.

Fig. 3: “Control” and “Metabolism” are not particularly helpful labels for the panels without an explanation of what those terms mean in the caption.

Lines 181-182; 184-185: What is the previous explanation for why a long sporogonic cycle guarantees a large number of sporozoites in the salivary glands? Does this not have anything to do with a long sporogonic cycle allowing for more production of oocysts?

Line 186-187: The wording is a bit confusing here, as increasing mosquito longevity should select for slower development, so just saying mosquito longevity increases selection for fast development is a bit odd. Also, you don’t show any simulations that show the effect of altering mosquito longevity on the length of the sporogonic cycle, anyway, so it seems a bit odd to state this as a major result. The fact that reducing host lifespan selects for fast development is also a classic result in evolutionary epidemiology, so not surprising at all.

Reviewer #2 (Remarks to the Author):

This paper develops an individual-based model of malaria infection in mosquito vectors and human hosts to explore the evolution of a key malaria parasite life history trait: the length of the sporogonic cycle, i.e., development time inside mosquitoes (or extrinsic incubation period). Conventional wisdom suggests that malaria parasites should be under strong selection for fast growth inside their mosquito vectors, since most mosquitoes don't live for very long, and parasite fitness is equated with onward transmission. By incorporating details of mosquito metabolism — where bloodmeals provide resources both for mosquitoes to produce eggs and for parasites to produce oocysts/sporozoitcs — the authors show that selection can actually favour slower parasite growth that takes advantage of resources accrued through multiple blood meals. The model, understandably, simplifies some of the components of a very complicated system, but in a way that allows the authors to focus on the key details under investigation. Overall, the authors provide a plausible explanation for why selection may not invariably favour faster development times for malaria parasites.

I think this paper provides an interesting perspective on an important trait of malaria parasites. In the interests of full transparency, I have previously reviewed this paper at another journal. At that time, I was fairly positive about the paper, but had some questions about model assumptions and inferences/interpretation. I am happy to see that the manuscript has been revised in ways that address a number of those questions.

Main comments.

1. One question I previously had that I don't think was addressed was what limits parasite fitness on the upper end of sporogonic development? In other words, why does $T_{sp}=12$ perform better than $T_{sp}=13$ in the metabolism model? The intuition here is that mosquito lifespans are driving this, but presumably that would be easy to show with some sort of sensitivity analysis (altering mosquito death rates up or down and seeing if the optimal T_{sp} changes as expected).
2. If I understand correctly how the percentages in Figure 1D are calculated, these will potentially be underestimates of the percent of spreaders or superspreaders at any given time in the simulations. My thinking is that because of the differences in average lifespans, and because all new mosquitoes start out uninfected, the numbers of uninfecteds (and carriers) will accumulate over the course of the simulations faster than the other two groups. So, dividing by the total number of mosquitoes that ever lived will give a lower % of spreaders/superspreaders than are actually part of the mosquito population at any given point in time. I'm sure that the spreaders and super spreaders will still be rare in this case, but I am curious how different the values would be if one calculated something like "average %" of each type, say from whenever the mosquito populations reach those steady states.
3. I think a bit more could be done to sign post that there are really two different models here (the IBM and the ODE) and the second one isn't explicitly embedded in the first. The line "we integrated the model of nutrient allocation into our individual-based model" (l.130) sort of suggests this explicit embedding, and I know the next sentence explains that this was done in a simplified way, but it takes a very close reading to appreciate that the insights gained from the ODE are captured in a more implicit or phenomenological way in the IBM. Around l. 130, I might describe this as "integrating the

effects/consequences of nutrient allocation into our individual-based model". I would suggest also laying this out a bit more clearly in the introduction, l. 45-54.

4. l.329-330 suggests that 2 bloodmeals achieves transmission probability equivalent to the the control IBM. But that doesn't seem right to me if $p_0=0.2$ (l. 328) and $p_{M \rightarrow H}$ in the control model = 0.7 (Table 1).

By my calculation, according to the Hill function defined on line 329:

- for one blood meal, $p_{M \rightarrow H} = 0.2 + 1/(1+1) = 0.7$
- for two blood meals, $p_{M \rightarrow H} = 0.2 + 2/(2+1) = \sim 0.87$

What am I missing here?

In any case, I assume the evolutionary results will be pretty sensitive to the function that translates bloodmeals to transmission probability, so it would be nice to see some sort of sensitivity analysis here too (presumably, most easily achieved by altering p_0). This could also serve as a stronger call for more future empirical work on this relationship!

5. I think it should be more clear from the title and/or abstract which trait of the malaria parasites is under study. "parasite development" is mentioned about halfway through the abstract, though it's not made precisely clear that we're talking about "development time" until a few sentences later. I'm also surprised that the expression "extrinsic incubation period" doesn't show up until the discussion (l. 207) when this seems like common nomenclature for this trait.

Minor points.

- l. 114-115, actually it looks like the energy accumulated decreases from the first to the second blood meal in the uninfected case (3BM) too. I'm not sure I understand why that would be, but perhaps equation 8 offers an explanation?
- l. 235, I'm confused by the phrase "Parasites are embedded in every host", since there are susceptible hosts (and vectors).
- l. 249, "occysts" → oocysts

We appreciate the reviewer's work in carefully evaluating our manuscript, which have contributed to improve its quality. Specifically, the recommendations regarding the manuscript's readability have guided us in reorganizing the content, resulting in a more accessible narrative.

The following pages list our detailed point-by-point responses to each of the comments of the reviewers. Revisions in the article are shown in blue font. We hope that the revisions in the manuscript and our accompanying responses will be sufficient to make our manuscript suitable for publication.

REVIEWER COMMENTS

Reviewer #1 (Remarks to the Author):

In this paper, the authors show how slow parasite development can benefit Plasmodium parasites in the mosquito host by allowing it to take advantage of the extra resources to developing oocysts provided by the host taking multiple blood meals, and that the balance between within-host selection for slow development and between-host selection for fast development can explain the observed duration of the sporogonic cycle. They do this by deriving a simplified, but mechanistic, model for both mosquito reproduction and Plasmodium development, and then nesting this model in an individual-based model of Plasmodium transmission among human and mosquito hosts. I thought this paper was really interesting, if a bit difficult to read at times. I am hoping my comments below will help to clarify my points of confusion and help highlight the key results that make this paper fascinating.

My major structural comment is that a significant improvement in this paper would result, in my opinion, from rearranging the model presentation. In particular, I think you should start with the model of oocyst development (Fig. 2). That allows you to move forward in your manuscript your really interesting and absolutely key result (Fig. 2D) that parasites can benefit from a long sporogonic cycle because it allows them to take advantage of multiple blood meals to increase their production of oocysts before switching to sporozoite production. I would move all of the results shown in Fig. 1 into the Supplemental Material because the results shown there are pretty intuitive. I think you can just say something like, "Although our within-vector model shows that parasites benefit from a long sporogonic cycle, in nature that is a risk, as hosts must live long enough to take multiple blood meals. Given that mosquito mortality is age-dependent, the mosquitoes face an important trade-off between a long sporogonic cycle maximizing within-vector fitness and it potentially jeopardizing transmission of sporozoites to the human intermediate host. In the Supplemental Material we use an individual-based model of transmission with a fixed sporogonic cycle to show that although most mosquitoes do get multiple blood meals, very few live long enough to take multiple blood meals after an infectious blood meal, and these are very important to overall transmission." Then, in the Supplemental Material, I would replace Fig. 1F with a figure that shows the average number of blood meals following an infectious blood meal to emphasize how rare it is to obtain multiple blood meals following the infectious one (as you do in Fig. 3C). This allows you to really simplify the model presentation because rather than explaining the individual-based model, explaining the within-host model, and

then explaining how to embed the within-host model into the individual-based model, you can just do the last two.

Thank you for this suggestion. We agree that starting with the ODE model of resource allocation makes the paper easier to read. We have restructured the order of the results presentation accordingly and moved the previous Figure 1 into the Supplementary Materials (Supplementary Figure 1). Additionally, we also moved the description of the individual based model in the Methods, and it appears now after the resource allocation ODE model.

Importantly, we have kept an improved description of the individual-based model in the Results since, as you correctly pointed out in your later comment, it is essential to understand the details of the following sections.

We hope the manuscript is now more accessible while containing all necessary details.

I also felt that the Discussion section did not do enough to place this new model within the broader modeling literature. Although malaria is not my area of research, I am familiar with the evolutionary epidemiology literature – the approach you are using here of combining within-host and between-host models is often called a “nested model” approach. A quick Google Scholar search for (“nested model” + *Plasmodium*) brings up 408 results, including some that you cite (such as Costa et al. 2018 Nature Communications, although that was not actually cited in the Discussion section) and many that you do not. No doubt most of those models are not relevant here, but including something about how your nested model differs from other nested models that have been used to study malaria evolution would be useful. Perhaps some of this is on lines 197-204, but it is unclear how those models were constructed, and I know that there are a lot of within-host models that do include relatively complex descriptions of the *Plasmodium* life cycle (e.g., Mideo et al. 2008 American Naturalist) – have any of these been nested into a between-host model?

Thank you for raising this point. There are indeed nested models in the malaria transmission literature, but they focus on the infection dynamics in the human host (Bushman et al. 2016 Royal Proceedings B). Only very few studies include *Plasmodium*'s within-vector development beyond the classical Ross-MacDonald model description (Costa 2018 Nature Communications, Childs 2017 PLoS One), but none of these models considers mosquito metabolic processes.

Importantly, parasite evolution has not been traditionally studied in any of the model extensions we mention in the Discussion, and only one study has investigated the effects of immune dynamics and drug resistance but in the human host (Whitlock et al. 2021 PLoS Comp Bio). Thus, to the best of our knowledge this is the first model to study the mosquito-driven evolution of *Plasmodium* parasites.

We have now explicitly highlighted the novelty of including mosquito metabolism into the epidemiological model in the Discussion in lines 235-245 as follows:

“Our study demonstrates the importance of conceptualizing complex parasite-vector interactions into models of *Plasmodium* transmission to realistically predict malaria epidemics and evolution. Since the first mathematical model developed by Ross and Macdonald (MacDonald 1956), there has been an expansion of theoretical approaches that consider a variety of geographical, ecological and epidemiological complexities

(reviewed in Reiner 2013, Smith 2014, Smith 2017), as well as multiple mosquito life-history traits (e.g., larval stages (Eckhoff 2011), biting frequency (Filipe 2007, Griffin 2010), feeding, and movement patterns (Pizzitutti 2015)). However, only very few of these models explicitly consider within-vector parasite development (Costa 2018, Childs 2017), beyond the original Ross-MacDonald description (Ohm 2018, Smith 2004, Reiner 2013, Smith 2018). Importantly, parasite evolution has not been traditionally studied in any of these model extensions, and only one study has investigated the effects of immune dynamics and drug resistance in the human host (Whitlock 2021). Here, we study the mosquito-driven evolution of *Plasmodium* parasites, and show that if mosquito metabolism is explicitly modeled, the exponential relationship between sporogonic cycle and mosquito life-span no longer holds (Figure 6). Thus, our results indicate that T_{sp} is not as sensitive in determining transmission intensity as was originally suggested (Shaw 2020). “

Specific comments:

Line 38-39: This is a really fascinating statement, and one I would like to see explained a bit more. When you say that the, “parasites rely on the same resources,” do you mean that very specifically, that they use essential yolk proteins for oocyst and sporozoite development? Also, HOW does reproductive investment and nutrient allocation influence oocyst and sporozoite development? Is that the more the mosquito invests in reproduction, the more parasite reproduction occurs? The next sentence suggests that it is only surplus metabolic resources that are available to the parasite, so it would seem possible that the amount of surplus is relatively unaffected by the total allocation to reproduction (e.g., if it takes 1 unit of resource to make an egg, and the mosquito allocates 4.5 units of resource to reproduction, then the mosquito will make 4 eggs with 0.5 units surplus; if it allocates 8.5 units of resource, then it makes 8 eggs and still has only 0.5 units of surplus). I would love to have a bit more explanation of what is going on here, and it seems like this process is quite critical to understanding the rest of the manuscript.

Thank you for raising this question. The metabolic relationship between malaria parasites and their mosquito hosts is indeed complex. We have included the missing basic information into the Introduction, in lines 34-46, as follows:

“*Anopheles* mosquitoes mostly feed on plant nectar, and only females take a blood meal to replenish their amino acid and lipid stores required for their reproductive cycle. In contrast to non-hemophagous insects like *Drosophila* that progressively accumulate resources for reproduction, mosquitoes have evolved a different strategy, characterized by a rapid nutrient assimilation from a blood meal. This process floods the mosquito with the essential nutrients for egg development within a short period of 48 hours, after which the reproductive investment is restrained [Lampe et al. 2019 Nature Communication]. *Plasmodium* parasites exploit the blood-feeding behavior as an entry point and also as nutritional source for their own development. The co-dependance on blood meal derived lipids for egg and parasite development has been reported to rely on nutrients/lipids carried by the lipid transporter lipophorin [Costa et al. 2018, Nature Communications, Werling et al. 2019, Cell]. Interestingly, the allocation of nutrients to reproduction inversely affects oocyst and sporozoite development. For example, increasing reproductive investment by prolonging the mosquito’s reproductive cycle, compromises *Plasmodium*’s sporogonic development. Consequently, limiting reproductive investment benefits parasite development [Lampe et al. 2019 Nature Communications]. Importantly, mosquitoes do not exhaust all available resources on reproduction [Lampe et al. 2019

Nature Communications], and therefore, the surplus metabolic resources that are available to parasites can vary from one mosquito to another.”

Line 43-44: Can you explain what you mean by, “How would malaria parasites adapt to the metabolic states of their mosquito host?” What is the evolving trait in this adaptation? Is it the relative allocation between growth and reproduction, as alluded to in the second paragraph?

We apologize for the confusion. The question is whether parasites would react to the different metabolic traits of the mosquitoes, i.e., whether their development would be affected in an environment with less metabolic resources. To avoid confusion, we have removed these questions.

Lines 60-83: I really struggled to understand the structure of the model. I realize that Nature Communications puts the methods at the end, but there needs to be enough explanation of the model in the Results to understand how the Results were derived, and that was not the case for me. For instance, you say that (line 61-62) that randomly chosen humans and mosquitos experience either birth, death, or infection. How do you decide which event occurs? Then the very next sentence says that mosquitos also grow and pupate – how does that fit into the “birth, infection, and death” of the previous sentence? How is growth, death, and pupation affected by ecological variables? Next, you go on to describe the developmental cycle of the Plasmodium, but you initially just encapsulate the entirety of mosquito development into the delay between exposure and infectiousness. Next, you include the process of recovery for the human population, which again does not fit into your statement that the stochastic processes are “birth, infection, and death.” Then you say that individuals that fail to recover become chronically infected, but with lower parasitemia (but not a lower probability of death, contradicting a previous statement that you assume infection increases probability of death because of a high level of parasitemia). But if recovery is a stochastic process, then any individual, at any time, could recover – how then do you decide when an individual has become chronically infected?

We apologize for the confusion in the model description. We have modified the Results section addressing specifically your concerns

Lines 110-115:

“In this model, human and mosquito individuals are randomly selected during every time step of one day and exposed to specific events with probabilities described in the Methods section. These events include birth, growth, infection, death, and recovery in the case of humans. Our model encompasses the complete mosquito life cycle, including larval and adult stages. Individual larvae exhibit variability in the time of pupation, growth rate, and death rate, which leads to population heterogeneity. We assume that all larval parameters (mentioned above) depend on ecological variables, such as density, temperature, and carrying capacity (see Methods). “

Lines 121-131:

“Infection occurs when a mosquito bites a *Plasmodium*-infected human. To simplify the representation of complex processes involved during parasite infection, we consider three mutually exclusive infection states in mosquitoes (susceptible, exposed, and infectious), and five in humans (susceptible, exposed, infectious, recovered, reservoir).

Moreover, parasites are defined only by their sporogonic cycle T_{sp} . Thus, upon an infectious bite, mosquitoes carry the parasite (*exposed*), but are *infectious* only after $T_{sp} = 13$ days. For simplicity, we consider that infectious mosquitoes transmit the parasite only to susceptible humans. We implement an incubation period in humans of $T_{IP} = 28$ days, after which exposed humans become *infectious*. During infection, humans have a higher death rate, simulating a high level of parasitemia $\delta_H = \delta_H + \delta_{p,H}$ where δ_H is the intrinsic human death rate, and $\delta_{p,H}$ the parasitemia. Only during the infectious state, humans can recover at a rate p_r , becoming immune to the parasite (*recovered*). Humans that fail to clear the infection become chronically infected, but harbor lower parasitemia ($0.001 \delta_{p,H}$), and a lower probability of transmitting the parasite ($0.1 p_{H \rightarrow M}$). We considered these individuals to be asymptomatic carriers, i.e., are in the *reservoir state*."

Line 63: Is this saying the traits are (1) the time of pupation; (2) growth rate; (3) death rate? Or are you saying the traits are the rates of pupation, growth, and death?

See detailed answer above, the traits are the time of pupation, growth rate, and death rate.

Line 77: What does the "EIP" in T_{EIP} stand for?

EIP stands for extrinsic incubation period, which is defined as the duration of parasite development in the mosquito. We acknowledge that this notation is incorrect, and have changed it to T_{IP} , standing for "incubation period" as defined also in the main text.

Fig. 1D: There is very little variation around the percentages, but if you add the percent of female mosquitoes in each category up, they look to be over 100% - the figure indicates that very close to 100% of females are uninfected. Maybe this is because the results are presented on a log scale - you might actually put in the caption what the average percent of females in each category is to avoid confusion.

This is a very good point. Because of the log scale, the results may be confusing. In reality 94.41 % are uninfected, 5.21 % carriers, 0.35 % spreaders, and 0.03% super-spreaders. We have included these numbers in the caption of the new Supplementary Figure 1.

Line 90-92: Why is this surprising? This seems completely intuitive since mosquitoes have to live long enough to become infected (which may be rare), then live long enough to become infectious (which takes at least 13 days), then live long enough to take a blood meal (and become a spreader), then live long enough to take a second blood meal (at least three more days to become a super-spreader).

Indeed, the results that older mosquitoes transmit malaria is intuitive. The most surprising finding was that only 0.1% of mosquitoes are sufficient to maintain stable epidemics (one order of magnitude lower compared to the prevalence of infected mosquitoes). We have removed the 'surprise' to avoid confusion.

Lines 96-116: Based on the structure of the within-host model of energy allocation in Fig. 2A, I do not see how it is possible for Plasmodium oocysts (O) to not compete with mosquito oocysts (E) for reserves (R) even after the first blood meal because (line 105) they take, "resources from the vector's reserves at a constant rate β_P ." Without an explicit delay, the Plasmodium will be constantly taking some energy from reserves; I assume this is happening, just at a very low rate compared the rate of allocation to eggs,

and that the total rate of energy consumption from reserves depends on the number of Plasmodium oocysts. That would make sense to me: after a single blood meal, there are a very low number of oocysts, so even though they are taking some reserves away from possible egg production, you don't see the impact; but when there are multiple blood meals, by the second blood meal O is large enough for the energy drain from reserves to impact mosquito reproduction. If that is what is happening, I would say that it is not that (line 112), "the parasite is not in direct competition with its vector." It is always in direct competition with its vector, but you can only see that competition when O is large enough to have an impact, which only occurs if there are multiple blood meals. To put it another way, if you assumed that the single infectious blood meal contained a very large number of developing oocysts, so $O(0)$ was appreciably large, I hypothesize that you would see an impact on mosquito reproduction.

Thank you for raising this important point. You are correct: in the model the total rate of energy acquisition grows with increasing amount of the metabolic energy inside the oocyst compartment. Therefore, if $O(0)$ would be very large, a strong competitive scenario would be seen during the first blood meal. It is important to recall that during the initial 48 hours post infection the incoming parasites only complete midgut invasion and start their transformation into young oocysts. The ookinetes and young oocysts are metabolically less active and require approximately 5-6 days before initiating active growth. We abstracted this process by having a very low $O(0)$, effectively modelling early metabolically inactive parasites. As a result, we observe different time scales of egg and parasite development that result in a very weak competition in the 1st blood meal like the one observed experimentally (Costa et al. 2018 Nature Communications).

We have corrected this sentence in the results section lines 81-83 of the resource allocation model description:

"Importantly, the ookinetes and young oocysts are less active metabolically and require approximately 5-6 days before initiating active growth (Costa et al. 2018 Nature Communications et al.). We model these early metabolically inactive parasites by having a very low initial metabolic energy in the oocyst compartment ($O(0) = 0.01$)."

And in lines 89-92:

"Because in our model the total rate of energy acquisition grows with increasing amount of metabolic energy inside the oocyst compartment, the early (and metabolically low energy) parasites cannot scavenge enough resources, resulting in a very weak competition with its vector, like that observed experimentally (Costa 2018 Nature Communications)."

Line 142-143: How do classical models incorporate T_{sp} ? I realize this is explained in detail in the Discussion, but a brief explanation along the lines of the caption of Fig. 6 might answer some questions for a reader here.

This indeed an important point. We have changed the caption as follows:
"By only considering basic life-history traits of mosquitoes (larval stages, biting behavior, reproduction, and death) and a simple within-vector *Plasmodium* development (an exponential relationship between mosquito longevity and transmission probability), classical models of malaria transmission predict that a reduction of the sporogony time (T_{sp}) increases parasite's fitness and aids transmission success."

Fig. 3: “Control” and “Metabolism” are not particularly helpful labels for the panels without an explanation of what those terms mean in the caption.

Thank you for pointing this out. We have included a definition of control and metabolism in the caption of new Figure 2 as follows:

“Summary N = 15 stochastic simulations excluding (control simulations, purple) or including the effects of mosquito metabolism (cyan).”

Lines 181-182; 184-185: What is the previous explanation for why a long sporogonic cycle guarantees a large number of sporozoites in the salivary glands? Does this not have anything to do with a long sporogonic cycle allowing for more production of oocysts?

We are sorry for the confusion. The number of oocysts is determined within the first 48 hours post infection by the ookinetes that traverse the midgut and remains constant throughout later parasite development. Oocysts do not rupture synchronously, therefore a few early sporozoites may be observed already 10 days post an infectious blood meal (Werling et al. 2019 Cell). However, only after 14 days the salivary glands will be invaded by a large number of sporozoites.

A large number of sporozoites is beneficial for transmission for two major reasons. First, there is a threshold in the number of sporozoites that allows transmission, i.e., many sporozoites need to be in the salivary glands to ensure that at least a few are injected into the host. Second, a large number of sporozoites damages the salivary glands and increases mosquito biting frequency.

These are the two major explanations that have been provided to explain why a long sporogonic cycle might be beneficial for *Plasmodium* parasites.

We have modified the Discussion lines 219-221:

“Based on the concept that long sporogony guarantees a large number of sporozoites in the salivary glands, two major explanations have been provided until now. First, a high number of sporozoites is necessary to transmit at least a few to the vertebrate host (Koella et al 1999, Schwartz 2001); and second, a high number of sporozoites stimulates mosquito biting to increase transmission (Koella et al 1999, Paul et al. 2003).”

Line 186-187: The wording is a bit confusing here, as increasing mosquito longevity should select for slower development, so just saying mosquito longevity increases selection for fast development is a bit odd. Also, you don't show any simulations that show the effect of altering mosquito longevity on the length of the sporogonic cycle, anyway, so it seems a bit odd to state this as a major result. The fact that reducing host lifespan selects for fast development is also a classic result in evolutionary epidemiology, so not surprising at all.

Thank you for commenting on this. We have changed the sentence to “mosquito life span” instead of mosquito longevity to avoid confusion in line 224. Moreover, this conclusion is now supported by additional simulations presented in our new Figure 5 a,

and described in the Results section, lines 187-197:

5: Mosquito longevity and parasite's strength determine the evolution of *Plasmodium's* sporogonic cycle. Evolved sporogonic development time (Tsp) at the end of the evolutionary simulations (at t = 10,000 days). **a** We simulated two additional mosquito populations with different age-dependent death rates by changing the shape parameter (x) of the Gompertz distribution, resulting in younger ($x = 0.18$) and older ($x = 0.09$) mosquitoes compared to our previous simulations. **b** We also modified the slope of the Hill function describing the mosquito-to-human transmission probability $p_{M \rightarrow H}(N_{BM}) = p_0 + \frac{n_{BM}}{n_{BM} + h}$, effectively modeling parasites with low ($h=2$), intermediate ($h=1$), or high ($h=0.5$) scavenging strength. Note that the simulations ran with parameters $x = 0.16$ and $h=1$ correspond to the results depicted in Figure 4. All simulations were started with a fast initial sporogonic development (Tsp(0) = 10 days). Box plots show the median with first and third quartile, whiskers depict min and max values. Summary of N=10 stochastic simulations.

“To investigate the influence of mosquito lifespan on *Plasmodium* evolution, we conducted two additional sets of simulations by adjusting the shape parameter x of the Gompertz distribution, effectively simulating mosquitoes with higher ($x = 0.089$ resulting in median survival of 27 days) and lower ($x = 0.18$ resulting in median survival of 17 days) survival probabilities compared to our current simulations ($x = 0.169$ resulting in median survival of 17 days). Note that modeling a larger decrease in mosquito life span would not assure stable infection dynamics. Therefore, we chose to model only a modest decrease in mosquito life span as a proof of principle.

Starting the simulation with a 'fast' Tsp = 10 days, we observed that malaria parasites evolved an even longer Tsp = 14 days when mosquitoes had an extended lifespan. Conversely, in mosquitoes with a shorter lifespan than in our previous simulations, the evolution of *Plasmodium* was limited already at Tsp = 11.5 days, confirming that mosquito lifespan constrains the benefits of an extended sporogony period. “

Reviewer #2 (Remarks to the Author):

This paper develops an individual-based model of malaria infection in mosquito vectors and human hosts to explore the evolution of a key malaria parasite life history trait: the

length of the sporogonic cycle, i.e., development time inside mosquitoes (or extrinsic incubation period). Conventional wisdom suggests that malaria parasites should be under strong selection for fast growth inside their mosquito vectors, since most mosquitoes don't live for very long, and parasite fitness is equated with onward transmission. By incorporating details of mosquito metabolism — where bloodmeals provide resources both for mosquitoes to produce eggs and for parasites to produce oocysts/sporozites — the authors show that selection can actually favour slower parasite growth that takes advantage of resources accrued through multiple blood meals. The model, understandably, simplifies some of the components of a very complicated system, but in a way that allows the authors to focus on the key details under investigation. Overall, the authors provide a plausible explanation for why selection may not invariably favour faster development times for malaria parasites.

I think this paper provides an interesting perspective on an important trait of malaria parasites. In the interests of full transparency, I have previously reviewed this paper at another journal. At that time, I was fairly positive about the paper, but had some questions about model assumptions and inferences/interpretation. I am happy to see that the manuscript has been revised in ways that address a number of those questions.

Main comments.

1. One question I previously had that I don't think was addressed was what limits parasite fitness on the upper end of sporogonic development? In other words, why does $T_{sp}=12$ perform better than $T_{sp}=13$ in the metabolism model? The intuition here is that mosquito lifespans are driving this, but presumably that would be easy to show with some sort of sensitivity analysis (altering mosquito death rates up or down and seeing if the optimal T_{sp} changes as expected).

Thank you for raising this very important point. As described in our detailed response to Reviewer 1, we have carried out additional simulations to test which aspects limit the parasite's fitness. Indeed, the optimal T_{sp} changes depending on the life span of the mosquitoes, shown in our new Figure 5 (see above).

In mosquito populations with a shorter life span, parasites evolve only to a T_{sp} of 11 days, while in simulations of mosquitoes with longer life span, the optimal T_{sp} would increase to 14 days. Note that in these simulations, the initial $T_{sp}(0) = 10$ days, thus there is always selection for longer sporogony periods, however the final T_{sp} is limited by mosquito life span.

2. If I understand correctly how the percentages in Figure 1D are calculated, these will potentially be underestimates of the percent of spreaders or superspreaders at any given time in the simulations. My thinking is that because of the differences in average lifespans, and because all new mosquitoes start out uninfected, the numbers of uninfecteds (and carriers) will accumulate over the course of the simulations faster than the other two groups. So, dividing by the total number of mosquitoes that ever lived will give a lower % of spreaders/superspreaders than are actually part of the mosquito population at any given point in time. I'm sure that the spreaders and super spreaders will still be rare in this case, but I am curious how different the values would be if one calculated something like "average %" of each type, say from whenever the mosquito populations reach those steady states.

Thank you for this comment. Given the stable infection dynamics, the proportion of uninfected vs. infected and infectious mosquitoes remains relatively constant throughout the entire simulation, as demonstrated in Supplementary Figure 3. Consequently, calculating the averages of uninfected, carrier, spreader, and super-spreaders for a specific time point, such as when the population reaches a steady state ($t = 750$ days), does not affect the proportions of the types, as shown in the Figure below (from 94.41% to 95.09% in uninfected, 5.21% to 4.56% in carriers, 0.35% to 0.3% in spreaders, and 0.03% to 0.08% in super spreaders).

This method of calculating the proportions of mosquito types mainly highlights the variability among simulations at a particular time point. However, this variability diminishes when calculating the average over the total number of mosquitoes that have ever lived.

3. I think a bit more could be done to sign post that there are really two different models here (the IBM and the ODE) and the second one isn't explicitly embedded in the first. The line "we integrated the model of nutrient allocation into our individual-based model" (l.130) sort of suggests this explicit embedding, and I know the next sentence explains that this was done in a simplified way, but it takes a very close reading to appreciate that the insights gained from the ODE are captured in a more implicit or phenomenological way in the IBM. Around l. 130, I might describe this as "integrating the effects/consequences of nutrient allocation into our individual-based model". I would suggest also laying this out a bit more clearly in the introduction, l. 45-54.

Thank you for this suggestion. We have clarified it in the Introduction (line 50):

"Here, we present a theoretical framework that integrates the effects of within-vector metabolism (i.e. the metabolically induced resource allocation initiated by blood feeding) into an individual-based model of malaria transmission"

And in the Results section (lines 143-144):

"To explore the impact of mosquito metabolism on *Plasmodium* transmission, we integrated the effects of nutrient allocation into our individual-based model."

4. l.329-330 suggests that 2 bloodmeals achieves transmission probability equivalent to

the the control IBM. But that doesn't seem right to me if $p_0=0.2$ (l. 328) and $p_{M \rightarrow H}$ in the control model = 0.7 (Table 1).

By my calculation, according to the Hill function defined on line 329:

- for one blood meal, $p_{M \rightarrow H} = 0.2 + 1/(1+1) = 0.7$

- for two blood meals, $p_{M \rightarrow H} = 0.2 + 2/(2+1) = \sim 0.87$

What am I missing here?

Thank for spotting the error. The text has now been changed to “1 bloodmeal achieves transmission probability equivalent to the control individual based model “ (line 365).

In any case, I assume the evolutionary results will be pretty sensitive to the function that translates bloodmeals to transmission probability, so it would be nice to see some sort of sensitivity analysis here too (presumably, most easily achieved by altering p_0). This could also serve as a stronger call for more future empirical work on this relationship!

Thank you for this suggestion. You are absolutely right. We examined the impact of the parasite's ability to scavenge resources by modifying the slope h of the Hill function that describes the mosquito-to-human transmission probability $p_{M \rightarrow H}(N_{BM}) = p_0 + \frac{n_{BM}}{n_{BM}+h}$ (see new Figure 5b). By changing the relation between number of blood meals and transmission potential, we could investigate how different parasites would evolve.

We have included the following results in the results section (lines 201-209):

“Consistent with our within-vector resource allocation model, we observed that parasites requiring more blood meals to increase their transmission probability ('low scavenging strength') do not benefit as much from long T_{sp} since they would risk exceeding the vector's lifespan. In contrast, parasites with a 'high scavenging strength', i.e., parasites that require fewer blood meals to increase transmission potential ($h=0.5$), benefit more from an extended T_{sp} , evolving a slightly longer 'optimal' $T_{sp} \sim 12,2$ days.

Notably, this effect reaches saturation due to the inherent characteristics of the Hill function, wherein the mosquito-to-human transmission probability readily approaches 1. Taken together, our simulations show that *Plasmodium's* evolutionary strategies in response to mosquito metabolism are shaped by the mosquito life span and the parasite's ability to effectively scavenge resources.“

And in the Discussion (lines 254-255):

“Our simulations show that the relationship between the number of blood meals and transmission is essential in determining *Plasmodium's* evolutionary outcomes, calling for more empirical work in quantifying this relationship”

Introducing of this finding has extended our results and strengthened the take home message. Thank you.

5. I think it should be more clear from the title and/or abstract which trait of the malaria parasites is under study. “parasite development” is mentioned about halfway through the abstract, though it's not made precisely clear that we're talking about “development time” until a few sentences later. I'm also surprised that the expression “extrinsic incubation period” doesn't show up until the discussion (l. 207) when this seems like common nomenclature for this trait.

Thank you for this suggestion, we included it in the abstract:

“Here, we examine the evolution of the time *Plasmodium* parasites requires to develop inside the vector (extrinsic incubation period) with a novel individual-based model of malaria transmission that includes mosquito metabolism.”

Similarly, we have defined T_{sp} and EIP to be the same in lines 79-80:

“The duration of this parasite development process is known as extrinsic incubation period (EIP) (hereafter called sporogonic cycle, T_{sp}), lasts approximately 10-14 days in natural systems.”

Minor points.

- l. 114-115, actually it looks like the energy accumulated decreases from the first to the second blood meal in the uninfected case (3BM) too. I'm not sure I understand why that would be, but perhaps equation 8 offers an explanation?

This is actually the effect of equation 5. The energy in the egg compartment (E) will grow proportionally to itself and to R , the energy in the Resource compartment. Therefore, the total amount of energy that will be stored in E , will depend on how much E and R is present at the time when new resources will be mobilized, i.e., when β_E is activated (equation 8). The initial conditions of R are slightly higher than when the second and third blood meal come in, resulting in a lower energy accumulated in the egg compartment.

- l. 235, I'm confused by the phrase “Parasites are embedded in every host”, since there are susceptible hosts (and vectors).

Thank you for this remark. We have corrected line 313 in the Methods:

“During infection, parasites are embedded in every host, and will be copied into a new host upon every transmission event.”

- l. 249, “occysts” → oocysts

Thank you, we have corrected the typo.

REVIEWERS' COMMENTS

Reviewer #1 (Remarks to the Author):

I am very pleased with the response to both my and the other reviewers concerns. I commend the authors for carefully considering and addressing all the points raised. I think this is an important and interesting study and would be happy to see it published in Nature Communications.

Reviewer #2 (Remarks to the Author):

This is a revised version of a paper I previously reviewed. I appreciate the responses of the authors and I think the changes they have made really strengthen the manuscript. In particular, I think the subtle reorganization of the results works really well. The different parts and how they fit together seemed more straightforward to follow now. Further, I'm super happy to see the analysis of what constrains T_{sp} at the high end, i.e., mosquito survival, parasite scavenging strength, and the new Figure 5.

I think this work is very cool and I'll be excited to see it in print.

A couple of very minor things:

- around l. 24 and the mention of work that looks at how parasites “adapt to changes in their host environment”. Greischar et al. 2019 Evolution 73: 2175 looks at the influence of mosquito demography and epidemiological dynamics on malaria parasite evolution. There is no interesting within mosquito stuff in that model, but it does get outside of the host!

- l. 141-142, I know this is described as “the conventional expectation”, but it feels like it could use a citation, maybe?

- l. 192, should it say 19 days for $x=0.169$? (It currently says 17 days for both “lower” and “current” survival rates.)

REVIEWER COMMENTS

We thank you the reviewer for this positive feedback.

to follow now. Further, I'm super happy to see the analysis of what constrains T_{sp} at

I think this work is very cool and I'll be excited to see it in print.

We thank the reviewer for this motivating comment and for all the past suggestions that guided us to this revised version.

- around l. 24 and the mention of work that looks at how parasites "adapt to changes in

Thank you the suggestion, we have now cited this study in the paragraph.

- l. 141-142, I know this is described as "the conventional expectation", but it feels like it

Thank you for the suggestion. We have cited the work of MacDonald (Macdonald, G. Epidemiological basis of malaria control. 1956.), in which mosquito longevity and biting rate are described as the important parameters of malaria epidemiology.

ays 17 days for both "lower" and "current" survival rates.)

Thank you for raising this point. We have corrected the text as follows:

"We conducted two additional sets of simulations by adjusting the shape parameter x of the Gompertz distribution to model mosquitoes with higher ($x = 0.089$ resulting in median survival of 27 days) and lower ($x = 0.18$ resulting in median survival of 17 days) survival rates compared to our current simulations ($x = 0.169$ resulting in median survival of 19 days"